# COVID-19 outcomes in patients taking cardioprotective medications

**Fritha J. Morrison[1], Maxwell Su[2,3], Alexander Turchin**[1,4]*

**1** Division of Endocrinology, Brigham and Women's Hospital, Boston, Massachusetts, United States of America, **2** Department of Biostatistics, Harvard T. H. Chan School of Public Health, Boston, Massachusetts, United States of America, **3** Phase V Technologies, Wellesley Hills, Massachusetts, United States of America, **4** Harvard Medical School, Boston, Massachusetts, United States of America

* aturchin@bwh.harvard.edu

**Data Availability Statement:** Data cannot be shared publicly secondary to policies of the institution that owns that data (Mass General Brigham). De-identified data are available for researchers who meet the criteria for access to

## Abstract

### Introduction

The coronavirus disease 2019 (COVID-19) caused a worldwide pandemic and has led to over five million deaths. Many cardiovascular risk factors (e.g. obesity or diabetes) are associated with an increased risk of adverse outcomes in COVID-19. On the other hand, it has been suggested that medications used to treat cardiometabolic conditions may have protective effects for patients with COVID-19.

### Objectives

To determine whether patients taking four classes of cardioprotective medications—aspirin, metformin, renin angiotensin aldosterone system inhibitors (RAASi) and statins–have a lower risk of adverse outcomes of COVID-19.

### Methods

We conducted a retrospective cohort study of primary care patients at a large integrated healthcare delivery system who had a positive COVID-19 test between March 2020 and March 2021. We compared outcomes of patients who were taking one of the study medications at the time of the COVID-19 test to patients who took a medication from the same class in the past (to minimize bias by indication). The following outcomes were compared: a) hospitalization; b) ICU admission; c) intubation; and d) death. Multivariable analysis was used to adjust for patient demographics and comorbidities.

### Results

Among 13,585 study patients, 1,970 (14.5%) were hospitalized; 763 (5.6%) were admitted to an ICU; 373 (2.8%) were intubated and 720 (5.3%) died. In bivariate analyses, patients taking metformin, RAASi and statins had lower risk of hospitalization, ICU admission and death. However, in multivariable analysis, only the lower risk of death remained statistically significant. Patients taking aspirin had a significantly higher risk of hospitalization in both bivariate and multivariable analyses.

confidential data. Requests for the de-identified dataset that contains all study variables can be sent to the Mass General Brigham Institutional Review Board at irb@partners.org. Data use agreement with Mass General Brigham will be required to receive the de-identified dataset. The authors did not receive any special privileges in accessing the data that other researchers would not have.

**Funding:** This research was funded in part by contract # ME-2019C1-15328 from Patient-Centered Outcomes Research Institute (http://www.pcori.org). The funder only provided financial support in the form of the authors' (FJM, MS, AT) salaries and research materials and did not play any role in study design, data collection and analysis, decision to publish or preparation of the manuscript. Phase V Technologies did not provide any financial support for the study and did not play any role in study design, data collection and analysis, decision to publish or preparation of the manuscript. The specific roles of the study authors are articulated in the 'author contributions' section.

**Competing interests:** AT has received research funding from Astra Zeneca, Edwards, Eli Lilly, Novo Nordisk and Sanofi; has equity in Brio Systems; and has served as a consultant for Covance and Proteomics International. MS is an employee of Phase V Technologies. This does not alter our adherence to PLOS ONE policies on sharing data and materials. None of the other authors have any competing interests.

## Conclusions

Cardioprotective medications were not associated with a consistent benefit in COVID-19. As vaccination and effective treatments are not yet universally accessible worldwide, research should continue to determine whether affordable and widely available medications could be utilized to decrease the risks of this disease.

## Introduction

The coronavirus disease 2019 (COVID-19) caused by infection with the SARS-CoV-2 virus continues to be a substantial threat worldwide. While often causing only mild symptoms, it can also lead to severe clinical outcomes and has resulted in over five million deaths worldwide as of December 2021 [1]. Individuals with atherosclerotic cardiovascular disease [2] and cardiovascular risk factors, such as diabetes, obesity, and chronic kidney disease, have elevated risk of adverse outcomes from COVID-19 [3–6]. Older patients are at a particularly high risk, and it has been postulated that this may be due to endothelial dysfunction and loss of endogenous cardioprotective mechanisms [7]. On the other hand, it has been proposed that some of the medications that reduce cardiovascular risk may also be beneficial for patients with COVID-19. HMG-CoA reductase inhibitors (statins) have anti-inflammatory and antithrombotic properties that may help mitigate disease severity in COVID-19 infection [8]. Renin angiotensin aldosterone system inhibitors (RAASi) also have anti-inflammatory properties and may block acute lung injury induced by coronaviruses and other viral infections [9–11], but their overall effect on outcomes of COVID-19 infection remains uncertain [12]. Metformin–a diabetes medication that is also thought to have independent cardioprotective effects [13]–also reduces inflammatory adipokines and TNFα which have been seen to contribute to COVID-19 severity [14]. Finally, aspirin has well-established anti-inflammatory and antiplatelet effects that may reduce risk of adverse outcomes [15]. We therefore conducted a study to examine the relationships between aspirin, metformin, RAASi, and/or statin use and the risk of adverse COVID-19 outcomes.

## Materials and methods

### Study design

We conducted a retrospective cohort study to examine the relationship between medications that reduce cardiovascular risk (aspirin, metformin, renin-angiotensin-aldosterone system (RAAS) inhibitors, and statins) and COVID-19 clinical outcomes.

### Study cohort

Our cohort was comprised of adults with COVID-19 (diagnosed based on a positive reverse transcription-polymerase chain reaction [RT-PCR] result for SARS-CoV-2) with at least one encounter with a primary care practice affiliated with Mass General Brigham prior to infection. Patients were included in the study if their first positive COVID-19 RT-PCR test was between the beginning of March 2020 (when regular screening for COVID-19 began in Massachusetts) and the end of March 2021 and were at least 18 years old at the time of positive test. Patients were excluded if their admission and discharge due to COVID-19 occurred before the first documented positive RT-PCR result for SARS-CoV-2 or if they had missing demographic

information. An individual patient served as the unit of analysis. If reinfection with COVID-19 was documented, only the first infection was studied.

This study was approved by the Mass General Brigham Institutional Review Board (protocol # 2020P003157). The requirement for informed consent was waived. Therefore no informed consent (written or verbal) was obtained from any participants in the study, as approved by the Institutional Review Board.

## Study measurements

Index date was defined as the date when the sample that tested positive for SARS-CoV-2 was taken (as opposed to the date when results were available). The primary outcome was hospitalization due to or associated with COVID-19 within 30 days of the first positive sample. Secondary outcomes included a) intensive care unit (ICU) admission; b) intubation during any hospitalization due to COVID-19 that started within 30 days of the first positive test; or c) death from any cause within 90 days of the first positive test.

Four classes of cardioprotective medications were assessed separately as predictor variables: a) aspirin, b) metformin, c) RAAS inhibitors, and d) statins. Anyone who had an active prescription for one of these medications on the date of their positive COVID-19 test was considered exposed, while those who had an indication for the medication class (defined as any previous prescription, but no active prescription for any medication in the class) were assigned to the comparison group. Prescriptions were considered active if the index date was within one year of the last prescription date and discontinuation was not documented prior to the index date. This led to four separate cohorts of patients: those with current or prior history of each cardioprotective medication class.

Confounders were obtained from the electronic medical records (EMR) at Mass General Brigham (MGB), an integrated health care system in New England. All variables were ascertained at the index date. Demographic characteristics included age in years, gender, race/ethnicity, marital status (partnered vs not), health insurance type (private vs other), median household income by zip code, and preferred language. Medical history variables were identified by any prior diagnosis code documentation of chronic lung disease, mental illness (dementia, bipolar disorder, schizophrenia, and schizoaffective disorder), and atherosclerotic cardiovascular disease (ASCVD, including coronary artery disease [CAD], cerebrovascular accident [CVA], and peripheral vascular disease [PVD]). Charlson Comorbidity Index (CCI), modified to exclude diabetes mellitus (DM), CAD, and chronic kidney disease (CKD), was calculated at the index date. In addition to diagnosis codes, elevated HbA1c $\geq$ 6.5% was used to identify patients with diabetes mellitus. Any documentation of a history of smoking, past or present, was categorized as history of smoking. Clinical measures from the index date or documented closest to and prior to index date were recorded for body mass index (BMI), systolic blood pressure (SBP), diastolic blood pressure (DBP), low density lipoprotein cholesterol (LDL), HbA1c, estimated glomerular filtration rate (eGFR), and proteinuria. eGFR was calculated using the CKD-EPI formula, excluding the adjustment for race [16].

## Statistical analysis

Summary statistics were calculated using frequencies and proportions for categorical data and means (SDs), medians, and ranges for continuous variables. Quantitative variables were compared using t-test and categorical variables using chi-square.

Multiple logistic regression models were constructed to estimate the associations between different classes of cardioprotective medications and poor COVID-19 categorical outcomes (hospitalization, ICU admission, intubation, and death). Previously described demographic,

clinical, and behavioral confounders were controlled for in the models. To attenuate the effects of outliers, eGFR was log transformed [17].

Secondary analyses were conducted using propensity scores as a covariate adjustment in logistic regression models of the treatment on each outcome. The propensity score represents a patient's probability of medication group assignment (current vs. previous use of specific cardioprotective medication class) and contains the information from all measured confounders. To estimate propensity scores, a non-parsimonious multiple logistic regression model was constructed with current cardioprotective medication as a dependent variable and potential confounding covariates as the independent variables [18]. Secondary analyses utilizing logistic regression models of each studied cardioprotective medication class on each outcome, while adjusting for propensity scores, were conducted to ensure consistency of our results.

Additional sensitivity analyses were conducted by limiting the comparison groups to a) patients with recent (18 months and 36 months) discontinuation of the study medications and b) documented history of adverse reactions to the study medications to help minimize unmeasured bias. Multiple imputation procedure with ten imputations was used to account for missing data (baseline HbA1c, SBP, DBP, BMI, LDL, and eGFR) both in multiple logistic regression analyses and propensity score analysis. For the propensity score analysis, propensity scores estimated from each imputed dataset were used individually to estimate treatment effects, which were combined to produce an overall estimate. All analyses were performed using SAS, version 9.4 (Cary, NC).

## Results

### Study cohort

A total of 13,585 patients with current or past prescriptions for the four cardioprotective medication classes studied were included in the analysis: 8,891 individuals with statin, 8,342 with RAASi, 4,487 with aspirin, and 3,696 with metformin use, past or present (Fig 1). Twenty-two percent (2,941 individuals) were not currently on any of the four medication classes at the index date, while 41.8% (5,683) were on one, 23.9% (3,241) on two, 10.6% (1,440) on three and 2.1% (280) on all four medication classes. Mean age of study patients was 62.5 years, ranging from 60.0 among those with previous metformin use to 67.7 among those with previous statin use (Table 1). More than half of study patients were obese, with mean BMI of 30.6 kg/m$^2$. Mean CCI scores were 2.2 after excluding diabetes, CKD and CAD from the calculation. Thirty-two percent of patients were not white, 16% identified their preferred language as one other than English, and females constituted 52% of the study population. Due to the timing of our case identification occurring during 2020 through March 2021, only 1% of patients had received at least one COVID-19 vaccination dose at their index date.

There were several differences in baseline characteristics between patients with past vs. current study medication use. Patients currently taking metformin, RAASi or statins had lower CCI compared to patients who were not (p < 0.0001 for all) while the opposite was true for patients currently taking aspirin (p = 0.0023). Past users were older than current users in the statin and RAASi groups and younger in the aspirin group. There were also differences in the distribution of patients over the study period: past study medication users were more likely to have had COVID-19 in Spring 2020.

Among all study patients, 1,970 (14.5%) were admitted to the hospital; 763 (5.6%) were admitted to an ICU; 373 (2.8%) were intubated and 720 (5.3%) died. Rates of adverse outcomes were much higher during the first COVID-19 surge in Spring 2020 than later in the study.

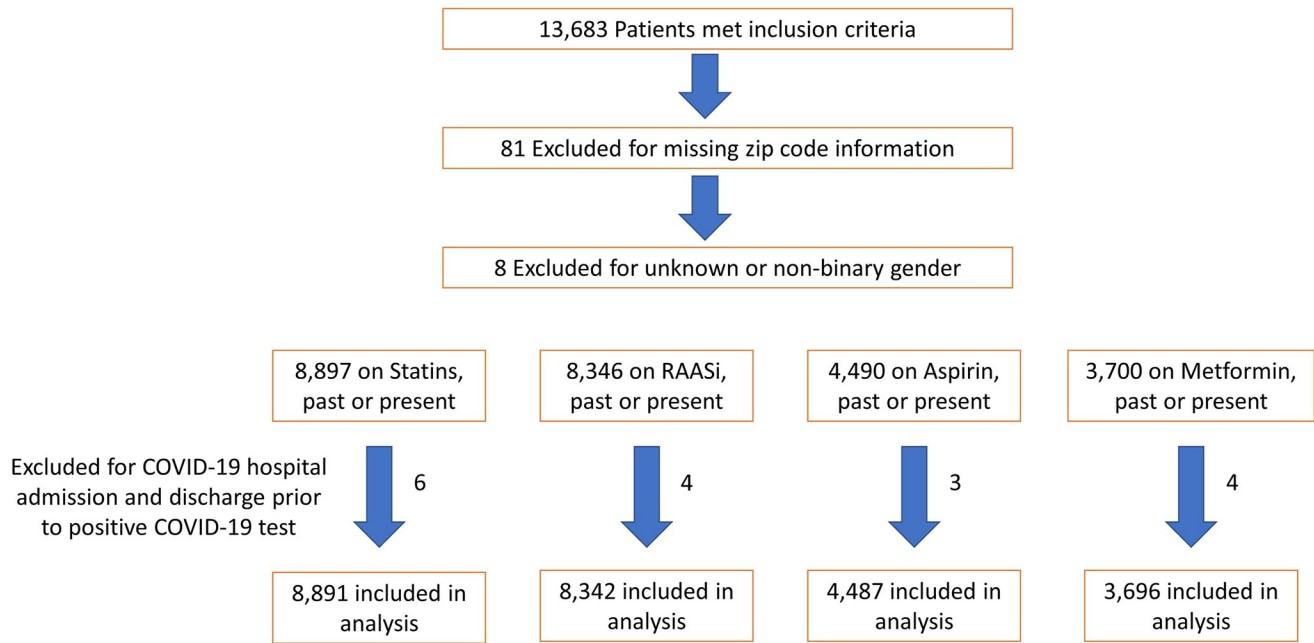

**Fig 1. Study patient flow.** Selection of study patients.

## Cardioprotective medications and COVID-19 outcomes

In bivariate analyses, patients taking statins and RAASi medications had lower risk of hospitalization, ICU admission and death (Fig 2). Metformin use was only associated with lower risk of death. Conversely, aspirin use was associated with increased risk of hospital and ICU admissions.

In multivariable analyses (Table 2), statin, RAASi and metformin use remained associated with a lower risk of death. Aspirin, on the other hand, was associated with an increased risk of hospitalization. Other patient characteristics that were associated with lower risk of adverse COVID-19 outcomes included female sex for all outcomes except mortality among metformin users, commercial insurance for hospitalization and mortality outcomes, and higher eGFR for mortality. Meanwhile, proteinuria and higher CCI scores were associated with higher risk of all adverse COVID-19 outcomes and elevated HbA1c was associated with higher risk of hospitalization in all groups and mortality in all groups, except the patients with current or past use of metformin. Higher BMI was associated with an elevated risk of hospitalization, ICU admission, and intubation across all groups. Lastly, psychotic disorders were associated with intubation and mortality for statin, RAASi, and aspirin users and only mortality in metformin users. Findings of the propensity score analyses (Table 3) and sensitivity analyses that limited the control groups to patients who had adverse reactions to study medications (Table 4) were also consistent with the primary analysis.

## Discussion

In this large study of over 10,000 patients, we for the first time examined both mortality and a range of other adverse short-term outcomes of COVID-19 for multiple classes of cardioprotective medications. The present study did not find evidence of consistent benefit of cardioprotective medications in patients with COVID-19. Many medications were associated with

**Table 1. Characteristics of study patients.**

| | Statin Population | | | | RAASi Population | | | | Aspirin Population | | | | Metformin Population | | | |
|---|---|---|---|---|---|---|---|---|---|---|---|---|---|---|---|---|
| | Current Statin Use | | Past Statin Use | | Current RAASi Use | | Past RAASi Use | | Current Aspirin Use | | Past Aspirin Use | | Current Metformin Use | | Past Metformin Use | |
| | Mean (SD) | N missing (%) | Mean (SD) | N missing (%) | Mean (SD) | N missing (%) | Mean (SD) | N missing (%) | Mean (SD) | N missing (%) | Mean (SD) | N missing (%) | Mean (SD) | N missing (%) | Mean (SD) | N missing (%) |
| Total, n | 7206 | | 1685 | | 6048 | | 2294 | | 1667 | | 2820 | | 2684 | | 1012 | |
| Age, years | 66 (12.5) | 0 (0) | 67.7 (15) | 0 (0) | 63.3 (14.2) | 0 (0) | 65.9 (16.9) | 0 (0) | 66 (15) | 0 (0) | 63.4 (16.7) | 0 (0) | 60.1 (13.8) | 0 (0) | 60 (16.8) | 0 (0) |
| Median household income by zip code, by $1,000 | 70.8 (24.9) | 0 (0) | 69.3 (25.9) | 0 (0) | 69.5 (24) | 0 (0) | 69.3 (25.7) | 0 (0) | 64.6 (24.3) | 0 (0) | 68.9 (25.5) | 0 (0) | 64.1 (22.6) | 0 (0) | 63.6 (22.1) | 0 (0) |
| HbA1c, %, | 6.5 (1.5) | 1174 (16.3) | 6.3 (1.4) | 265 (15.7) | 6.5 (1.5) | 1078 (17.8) | 6.3 (1.5) | 417 (18.2) | 6.6 (1.7) | 215 (12.9) | 6.1 (1.3) | 468 (16.6) | 7.5 (1.7) | 41 (1.5) | 7.1 (1.9) | 25 (2.5) |
| BMI, kg/m$^2$ | 30.6 (6.3) | 30 (0.4) | 29.8 (6.8) | 6 (0.4) | 31.5 (6.8) | 18 (0.3) | 29.8 (6.9) | 15 (0.7) | 30 (6.7) | 5 (0.3) | 30 (6.6) | 16 (0.6) | 33 (7.1) | 14 (0.5) | 32.8 (7.5) | 3 (0.3) |
| SBP, mmHg | 130 (17) | 24 (0.3) | 129 (18) | 7 (0.4) | 132 (17) | 24 (0.4) | 130 (18) | 20 (0.9) | 129 (18) | 10 (0.6) | 129 (17) | 17 (0.6) | 130 (16) | 9 (0.3) | 128 (18) | 7 (0.7) |
| DBP, mmHg | 75 (10) | 24 (0.3) | 75 (11) | 7 (0.4) | 77 (11) | 24 (0.4) | 75 (11) | 20 (0.9) | 74 (10) | 10 (0.6) | 75 (10) | 17 (0.6) | 76 (10) | 9 (0.3) | 75 (11) | 7 (0.7) |
| LDL, mg/dL, mean (SD) | 86 (36) | 217 (3) | 105 (45) | 70 (4) | 91.1 (36) | 285 (5) | 87.7 (36) | 152 (7) | 81 (35) | 104 (6) | 91 (36) | 154 (6) | 83 (36) | 73 (3) | 88.7 (38) | 38 (4) |
| eGFR, mL/min/1.73 m$^2$ | 73.8 (21.9) | 90 (1.2) | 70.8 (24.7) | 11 (0.7) | 75.9 (21.8) | 84 (1.4) | 69.6 (27.5) | 28 (1.2) | 72 (25.3) | 12 (0.7) | 75.4 (25.1) | 32 (1.1) | 81.6 (21.7) | 21 (0.8) | 76.7 (28.5) | 7 (0.7) |
| CCI | 2.4 (2.7) | 0 (0) | 2.9 (3) | 0 (0) | 2.2 (2.6) | 0 (0) | 3.1 (3.1) | 0 (0) | 2.9 (2.9) | 0 (0) | 2.7 (2.9) | 0 (0) | 1.9 (2.4) | 0 (0) | 2.5 (2.7) | 0 (0) |
| White, n (%) | 5176 (71.8) | | 1143 (67.8) | | 4125 (68.2) | | 1540 (67.1) | | 949 (56.9) | | 1833 (65) | | 1392 (51.9) | | 576 (56.9) | |
| Female, n (%) | 3206 (44.5) | | 915 (54.3) | | 2958 (48.9) | | 1307 (57) | | 801 (48.1) | | 1583 (56.1) | | 1368 (51) | | 640 (63.2) | |
| Partnered, n (%) | 4286 (59.5) | | 828 (49.1) | | 3447 (57) | | 1096 (47.8) | | 759 (45.5) | | 1473 (52.2) | | 1477 (55) | | 461 (45.6) | |
| Commercial Insurance, n (%) | 3517 (48.8) | | 681 (40.4) | | 3257 (53.9) | | 959 (41.8) | | 644 (38.6) | | 1300 (46.1) | | 1388 (51.7) | | 465 (45.9) | |
| ASCVD Baseline, n (%) | 2109 (29.3) | | 483 (28.7) | | 1326 (21.9) | | 702 (30.6) | | 732 (43.9) | | 750 (26.6) | | 536 (20) | | 251 (24.8) | |
| Diabetes Mellitus Baseline, n (%) | 2784 (38.6) | | 574 (34.1) | | 2267 (37.5) | | 798 (34.8) | | 725 (43.5) | | 786 (27.9) | | 2344 (87.3) | | 737 (72.8) | |
| Proteinuria, n(%) | 1250 (17.3) | | 388 (23) | | 1019 (16.8) | | 604 (26.3) | | 396 (23.8) | | 564 (20) | | 548 (20.4) | | 241 (23.8) | |
| Statin Meds, n(%) | 7206 (100) | | 0 (0) | | 3472 (57.4) | | 887 (38.7) | | 1137 (68.2) | | 1217 (43.2) | | 1863 (69.4) | | 412 (40.7) | |
| RAASi Meds, n(%) | 3472 (48.2) | | 445 (26.4) | | 6048 (100) | | 0 (0) | | 734 (44) | | 921 (32.7) | | 1569 (58.5) | | 344 (34) | |
| Aspirin Meds, n(%) | 1137 (15.8) | | 170 (10.1) | | 734 (12.1) | | 293 (12.8) | | 1667 (100) | | 0 (0) | | 466 (17.4) | | 121 (12) | |
| Metformin Meds, n(%) | 1863 (25.9) | | 190 (11.3) | | 1569 (25.9) | | 294 (12.8) | | 466 (28) | | 400 (14.2) | | 2684 (100) | | 0 (0) | |
| Dementia, n (%) | 112 (1.6) | | 50 (3) | | 68 (1.1) | | 61 (2.7) | | 44 (2.6) | | 57 (2) | | 28 (1) | | 16 (1.6) | |

*(Continued)*

**Table 1.** (Continued)

| | Statin Population | | | | RAASi Population | | | | Aspirin Population | | | | Metformin Population | | | |
|---|---|---|---|---|---|---|---|---|---|---|---|---|---|---|---|---|
| | Current Statin Use | | Past Statin Use | | Current RAASi Use | | Past RAASi Use | | Current Aspirin Use | | Past Aspirin Use | | Current Metformin Use | | Past Metformin Use | |
| | Mean (SD) | N missing (%) | Mean (SD) | N missing (%) | Mean (SD) | N missing (%) | Mean (SD) | N missing (%) | Mean (SD) | N missing (%) | Mean (SD) | N missing (%) | Mean (SD) | N missing (%) | Mean (SD) | N missing (%) |
| Psychotic Disorders, n (%) | | 76 (1.1) | | 19 (1.1) | | 46 (0.8) | | 28 (1.2) | | 34 (2) | | 30 (1.1) | | 38 (1.4) | | 28 (2.8) |
| Chronic Lung Disease, n (%) | | 294 (4.1) | | 74 (4.4) | | 195 (3.2) | | 115 (5) | | 79 (4.7) | | 119 (4.2) | | 100 (3.7) | | 38 (3.8) |
| Ever Smoked, n (%) | | 3620 (50.2) | | 857 (50.9) | | 2759 (45.6) | | 1127 (49.1) | | 838 (50.3) | | 1338 (47.4) | | 1195 (44.5) | | 448 (44.3) |
| Season | | | | | | | | | | | | | | | | |
| Spring 2020 | | 1603 (22.2) | | 491 (29.1) | | 1341 (22.2) | | 630 (27.5) | | 515 (30.9) | | 679 (24.1) | | 702 (26.2) | | 276 (27.3) |
| Summer 2020 | | 273 (3.8) | | 64 (3.8) | | 217 (3.6) | | 100 (4.4) | | 80 (4.8) | | 131 (4.6) | | 111 (4.1) | | 30 (3) |
| Fall 2020 | | 1071 (14.9) | | 263 (15.6) | | 931 (15.4) | | 326 (14.2) | | 219 (13.1) | | 442 (15.7) | | 418 (15.6) | | 146 (14.4) |
| Winter 2020 | | 4029 (55.9) | | 818 (48.5) | | 3365 (55.6) | | 1171 (51) | | 792 (47.5) | | 1455 (51.6) | | 1357 (50.6) | | 536 (53) |
| Spring 2021 | | 230 (3.2) | | 49 (2.9) | | 194 (3.2) | | 67 (2.9) | | 61 (3.7) | | 113 (4) | | 96 (3.6) | | 24 (2.4) |
| Vaccination Status | | | | | | | | | | | | | | | | |
| None | | 7118 (98.8) | | 1671 (99.2) | | 5974 (98.8) | | 2262 (98.6) | | 1645 (98.7) | | 2784 (98.7) | | 2653 (98.8) | | 1001 (98.9) |
| Partially | | 74 (1) | | 14 (0.8) | | 61 (1) | | 29 (1.3) | | 19 (1.1) | | 30 (1.1) | | 28 (1) | | 9 (0.9) |
| Fully | | 14 (0.2) | | | | 13 (0.2) | | 3 (0.1) | | 3 (0.2) | | 6 (0.2) | | 3 (0.1) | | 2 (0.2) |

Abbreviations: ASCVD, atherosclerotic cardiovascular disease; BMI, body mass index; CCI, Charlson comorbidity index; DBP, diastolic blood pressure; eGFR, estimated glomerular filtration rate; HbA1c, hemoglobin A1c; LDL, low density lipoprotein cholesterol; RAASi, renin angiotensin aldosterone system inhibitor; SBP, systolic blood pressure

mortality benefit but not a decrease in hospitalization, ICU admission, or ventilator support. On the other hand, treatment with aspirin was associated with a possible increased risk of hospitalization, but not other intermediate COVID-19 outcomes nor all-cause mortality. This study is a novel examination of the previously observed associations between cardioprotective medications and mortality because of its more complete picture of the proposed relationship by also looking at the associations between different medication classes and intermediate COVID-19 outcomes.

Our findings were broadly consistent with previously published literature [8, 14, 19–28]. Previous explorations of statin use generally found a mortality benefit [8, 23, 24, 26–28] but no change in the risk of ventilator use [8, 23, 27, 29, 30] or ICU admissions [23, 27, 30]. One study found that statin use was associated with lower mortality among patients with type 2 diabetes (T2DM) but not type 1 diabetes (T1DM) [26]. Several studies that found an increased risk [29] of or no association [28, 30] with adverse COVID-19 outcomes associated with statin use were limited to hospitalized patients [28–30] or had small sample sizes [19, 25]. Only one study that included but 58 patients with a plasma-cell disorder and COVID-19 found harm associated with statin use [19]. Most studies that found no association between RAASi use and COVID-19 mortality were small [19, 24, 31]. In the two larger studies showing no mortality benefit,

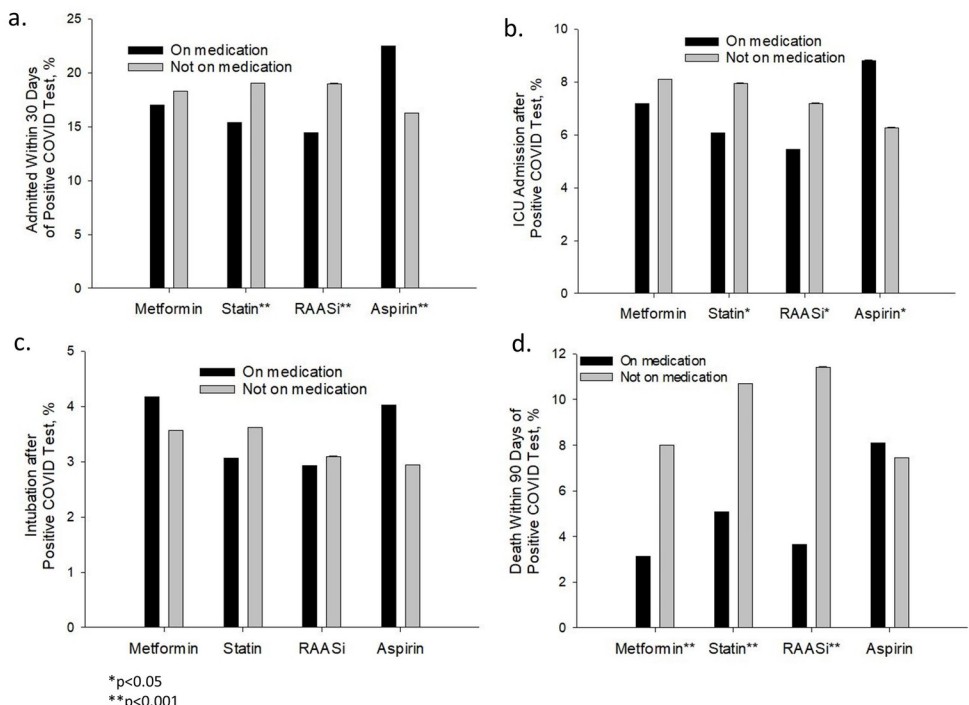

**Fig 2. Cardioprotective medications and study outcomes.** Incidence of COVID-19 outcomes in study patients.

Chen *et al.* only examined a hospitalized population [20] and Reynolds *et al.* were looking for a pre-specified 10% difference in outcomes rather than solely statistical significance [32]. Notably, a recent meta-analysis of clinical trials of angiotensin converting enzyme (ACE) inhibitors and angiotensin receptor blockers (ARBs) showed a mortality benefit that was limited to ARBs [33]. Previous literature on aspirin found no association between aspirin use and hospitalizations [19] or mortality [19, 24] in studies of both hospitalized patients and in a population of COVID-19 patients with plasma cell disorders. Chow *et al.* found aspirin to be linked to a reduction in ICU admissions, ventilator use, and mortality, but had a small sample size [34] and was limited to hospitalized patients. Lastly, our results were consistent with some of the largest previous studies of metformin and COVID-19 outcomes, which showed a reduction in mortality [4, 14, 21, 35]. Three other studies did not find an association between metformin use and mortality, but two were very small, with only 120 [20] and 58 [19] participants. Do *et al.* did not find an association between metformin vs no medication and mortality but did find a benefit of metformin over other non-metformin diabetes medications in a population of patients with T2DM [36]. Studies of metformin largely only examined mortality but no other outcomes.

Several possible mechanisms could have accounted for the improvements in COVID-19 outcomes observed in this study. Zhang *et al.* found an association between statin use and decreased risk of ventilator use and mortality [23]. They also found lower levels of inflammatory markers (C-reactive protein, interleukin 6 and neutrophil count) among statin users compared to non-users in their study, supporting a suggested anti-inflammatory mechanism for the decreased risk of poor COVID-19 outcomes among statin users [8]. Another possible mechanism for the lower risk of mortality among statin users is the observed association seen in De Spiegeleer *et al.*'s study: patients taking a statin were more likely to have asymptomatic

**Table 2. Effect of study medications and patient characteristics on COVID-19 outcomes: Multivariable analysis.**

a. Patients with Current or Past Statin Use

| | Hospitalization | | ICU admission | | Intubation | | Death | |
|---|---|---|---|---|---|---|---|---|
| | OR (95% CI) | P-value | OR (95% CI) | P-value | OR (95% CI) | P-value | OR (95% CI) | P-value |
| Current statin use | 0.935 (0.801, 1.091) | 0.3912 | 0.909 (0.729, 1.134) | 0.3986 | 0.923 (0.675, 1.262) | 0.6162 | 0.678 (0.539, 0.852) | 0.0009 |
| Current RAASi use | 0.96 (0.844, 1.091) | 0.5291 | 0.918 (0.761, 1.108) | 0.3732 | 0.899 (0.692, 1.166) | 0.4221 | 0.739 (0.596, 0.917) | 0.0059 |
| Current aspirin use | 1.295 (1.106, 1.517) | 0.0014 | 1.104 (0.88, 1.386) | 0.393 | 1.021 (0.743, 1.403) | 0.8971 | 1.125 (0.874, 1.447) | 0.3601 |
| Current metformin use | 1.149 (0.957, 1.38) | 0.1362 | 1.248 (0.965, 1.615) | 0.0911 | 1.428 (1.009, 2.021) | 0.0443 | 0.767 (0.556, 1.058) | 0.1059 |
| Age | 1.032 (1.025, 1.039) | < .0001 | 1.023 (1.013, 1.032) | < .0001 | 1.013 (1, 1.027) | 0.0489 | 1.073 (1.061, 1.085) | < .0001 |
| Female | 0.802 (0.705, 0.912) | 0.0007 | 0.738 (0.611, 0.891) | 0.0015 | 0.584 (0.448, 0.763) | 0.0001 | 0.531 (0.429, 0.659) | < .0001 |
| English | 0.924 (0.774, 1.102) | 0.3781 | 0.694 (0.544, 0.886) | 0.0033 | 0.722 (0.52, 1.002) | 0.0512 | 1.224 (0.898, 1.669) | 0.2006 |
| White | 0.775 (0.659, 0.913) | 0.0022 | 1.135 (0.895, 1.439) | 0.2974 | 0.781 (0.568, 1.073) | 0.1267 | 1.059 (0.794, 1.413) | 0.6947 |
| Partnered | 0.792 (0.698, 0.899) | 0.0003 | 0.898 (0.746, 1.08) | 0.2542 | 1.19 (0.919, 1.54) | 0.1879 | 0.941 (0.763, 1.162) | 0.5734 |
| Median Household Income By $1000 | 0.996 (0.993, 0.998) | 0.0014 | 0.986 (0.981, 0.99) | < .0001 | 0.994 (0.988, 0.999) | 0.0293 | 0.997 (0.993, 1.001) | 0.0996 |
| Commercial Insurance | 0.781 (0.684, 0.891) | 0.0003 | 0.658 (0.54, 0.802) | < .0001 | 0.592 (0.449, 0.781) | 0.0002 | 0.585 (0.458, 0.747) | < .0001 |
| History of smoking | 1.08 (0.952, 1.225) | 0.2304 | 1.22 (1.013, 1.469) | 0.036 | 1.102 (0.852, 1.426) | 0.4594 | 1.2 (0.973, 1.481) | 0.0882 |
| HbA1c | 1.103 (1.049, 1.161) | 0.0001 | 1.052 (0.98, 1.129) | 0.1624 | 1.093 (0.996, 1.198) | 0.0597 | 1.193 (1.095, 1.299) | 0.0001 |
| BMI by 10 kg/m² | 1.232 (1.117, 1.359) | < .0001 | 1.296 (1.129, 1.488) | 0.0002 | 1.758 (1.478, 2.091) | < .0001 | 0.924 (0.772, 1.106) | 0.3912 |
| SBP by 10 mm Hg | 0.993 (0.953, 1.035) | 0.7391 | 0.973 (0.917, 1.033) | 0.3747 | 1.045 (0.964, 1.134) | 0.2869 | 0.906 (0.849, 0.967) | 0.003 |
| DBP by 10 mm Hg | 0.977 (0.907, 1.054) | 0.5523 | 1.053 (0.943, 1.175) | 0.3568 | 0.939 (0.808, 1.092) | 0.4144 | 1.013 (0.895, 1.147) | 0.8405 |
| LDL by 10 mg/dL | 1.014 (0.995, 1.033) | 0.1457 | 1.037 (1.011, 1.065) | 0.0061 | 1.02 (0.983, 1.057) | 0.2904 | 0.983 (0.951, 1.016) | 0.3129 |
| eGFR[1] | 0.641 (0.559, 0.735) | < .0001 | 0.755 (0.625, 0.912) | 0.0035 | 0.779 (0.604, 1.004) | 0.0533 | 0.616 (0.506, 0.751) | < .0001 |
| Proteinuria | 1.764 (1.53, 2.035) | < .0001 | 1.808 (1.478, 2.213) | < .0001 | 1.973 (1.499, 2.598) | < .0001 | 1.593 (1.282, 1.98) | < .0001 |
| CCI | 1.059 (1.033, 1.086) | < .0001 | 1.062 (1.025, 1.102) | 0.0011 | 1.073 (1.02, 1.13) | 0.0069 | 1.129 (1.087, 1.173) | < .0001 |
| ASCVD | 1.035 (0.9, 1.19) | 0.6342 | 1.181 (0.964, 1.446) | 0.1083 | 1.071 (0.805, 1.425) | 0.6377 | 1.303 (1.054, 1.611) | 0.0145 |
| Chronic Lung Disease | 1.482 (1.142, 1.923) | 0.0031 | 1.111 (0.755, 1.637) | 0.5926 | 1.494 (0.914, 2.442) | 0.1094 | 1.351 (0.91, 2.006) | 0.1351 |
| Diabetes Mellitus | 0.997 (0.831, 1.196) | 0.9727 | 1.188 (0.917, 1.54) | 0.1923 | 0.987 (0.688, 1.416) | 0.9429 | 1.191 (0.899, 1.577) | 0.2224 |
| Dementia | 0.994 (0.682, 1.449) | 0.9769 | 0.91 (0.519, 1.596) | 0.7433 | 0.38 (0.118, 1.219) | 0.1037 | 1.181 (0.747, 1.866) | 0.477 |
| Psychotic Disorders | 1.63 (0.999, 2.66) | 0.0503 | 1.646 (0.864, 3.133) | 0.1294 | 1.594 (0.666, 3.812) | 0.2948 | 2.342 (1.115, 4.92) | 0.0247 |
| Season[2] | | | | | | | | |
| Spring 2020 | 1.298 (1.126, 1.496) | 0.0003 | 1.42 (1.158, 1.741) | 0.0007 | 1.711 (1.303, 2.246) | 0.0001 | 2.642 (2.124, 3.286) | < .0001 |
| Summer 2020 | 0.817 (0.584, 1.142) | 0.2362 | 0.867 (0.524, 1.434) | 0.5779 | 0.837 (0.401, 1.745) | 0.6347 | 1.512 (0.925, 2.473) | 0.0995 |
| Fall 2020 | 1.12 (0.936, 1.341) | 0.2169 | 1.143 (0.875, 1.495) | 0.3271 | 0.88 (0.584, 1.326) | 0.5411 | 0.994 (0.699, 1.414) | 0.9748 |
| Spring 2021 | 0.977 (0.679, 1.406) | 0.8999 | 1.323 (0.81, 2.161) | 0.2635 | 1.051 (0.502, 2.2) | 0.8957 | 0.813 (0.391, 1.694) | 0.581 |

b. Patients with Current or Past RAASi use

| | Hospitalization | | ICU admission | | Intubation | | Death | |
|---|---|---|---|---|---|---|---|---|
| | OR (95% CI) | P-value | OR (95% CI) | P-value | OR (95% CI) | P-value | OR (95% CI) | P-value |
| Current statin use | 0.896 (0.776, 1.036) | 0.1374 | 0.964 (0.777, 1.197) | 0.7405 | 0.942 (0.699, 1.269) | 0.694 | 0.778 (0.615, 0.985) | 0.0374 |
| Current RAASi use | 0.993 (0.858, 1.149) | 0.9264 | 0.969 (0.779, 1.205) | 0.776 | 1.119 (0.819, 1.527) | 0.4799 | 0.594 (0.476, 0.741) | < .0001 |
| Current aspirin use | 1.196 (1.001, 1.43) | 0.0489 | 0.989 (0.758, 1.29) | 0.9332 | 0.877 (0.602, 1.277) | 0.4929 | 1.13 (0.85, 1.504) | 0.3993 |
| Current metformin use | 1.227 (1.013, 1.486) | 0.0361 | 1.245 (0.946, 1.638) | 0.1171 | 1.632 (1.125, 2.367) | 0.0099 | 0.772 (0.549, 1.084) | 0.1349 |
| Age | 1.034 (1.027, 1.041) | < .0001 | 1.032 (1.022, 1.042) | < .0001 | 1.018 (1.004, 1.031) | 0.0088 | 1.072 (1.06, 1.084) | < .0001 |
| Female | 0.867 (0.758, 0.991) | 0.0371 | 0.713 (0.582, 0.874) | 0.0011 | 0.601 (0.454, 0.796) | 0.0004 | 0.586 (0.467, 0.736) | < .0001 |
| English | 0.964 (0.801, 1.159) | 0.6947 | 0.795 (0.609, 1.037) | 0.0902 | 0.958 (0.665, 1.381) | 0.8195 | 1.239 (0.888, 1.728) | 0.2065 |
| White | 0.801 (0.678, 0.947) | 0.0094 | 1.163 (0.905, 1.496) | 0.2385 | 0.768 (0.55, 1.07) | 0.119 | 1.119 (0.828, 1.511) | 0.4645 |
| Partnered | 0.852 (0.746, 0.972) | 0.0171 | 0.942 (0.772, 1.149) | 0.5554 | 1.166 (0.887, 1.533) | 0.2714 | 0.916 (0.733, 1.144) | 0.4384 |
| Median Household Income By $1000 | 0.995 (0.992, 0.998) | 0.0007 | 0.985 (0.98, 0.99) | < .0001 | 0.991 (0.985, 0.998) | 0.0084 | 0.998 (0.994, 1.003) | 0.4342 |
| Commercial Insurance | 0.795 (0.693, 0.913) | 0.0011 | 0.847 (0.688, 1.041) | 0.1151 | 0.785 (0.591, 1.042) | 0.094 | 0.631 (0.49, 0.812) | 0.0004 |
| History of smoking | 1.057 (0.927, 1.207) | 0.4071 | 1.047 (0.859, 1.278) | 0.6469 | 1.014 (0.77, 1.335) | 0.9219 | 1.215 (0.974, 1.515) | 0.0844 |

(*Continued*)

**Table 2.** (Continued)

| | Hospitalization | | ICU admission | | Intubation | | Death | |
|---|---|---|---|---|---|---|---|---|
| HbA1c | 1.103 (1.044, 1.165) | 0.0005 | 1.065 (0.986, 1.151) | 0.1092 | 1.133 (1.029, 1.248) | 0.0109 | 1.197 (1.09, 1.315) | 0.0002 |
| BMI by 10 kg/m$^2$ | 1.251 (1.134, 1.379) | < .0001 | 1.308 (1.132, 1.51) | 0.0003 | 1.813 (1.517, 2.166) | < .0001 | 0.948 (0.79, 1.138) | 0.5676 |
| SBP by 10 mm Hg | 1.02 (0.978, 1.063) | 0.3561 | 0.999 (0.939, 1.062) | 0.9662 | 1.057 (0.972, 1.148) | 0.1951 | 0.898 (0.839, 0.962) | 0.0021 |
| DBP by 10 mm Hg | 1.004 (0.93, 1.083) | 0.9245 | 1.132 (1.01, 1.269) | 0.0326 | 1.011 (0.865, 1.181) | 0.8909 | 1.023 (0.901, 1.162) | 0.7216 |
| LDL by 10 mg/dL | 1.002 (0.982, 1.023) | 0.8356 | 1.027 (0.997, 1.058) | 0.0803 | 1.021 (0.98, 1.063) | 0.3242 | 0.992 (0.958, 1.027) | 0.6371 |
| eGFR[1] | 0.676 (0.586, 0.78) | < .0001 | 0.749 (0.612, 0.916) | 0.0049 | 0.667 (0.511, 0.869) | 0.0027 | 0.579 (0.472, 0.711) | < .0001 |
| Proteinuria | 2.031 (1.757, 2.347) | < .0001 | 2.048 (1.661, 2.524) | < .0001 | 2.166 (1.629, 2.88) | < .0001 | 1.532 (1.217, 1.927) | 0.0003 |
| CCI | 1.063 (1.036, 1.091) | < .0001 | 1.071 (1.03, 1.113) | 0.0006 | 1.097 (1.037, 1.159) | 0.0012 | 1.085 (1.042, 1.129) | 0.0001 |
| ASCVD | 0.956 (0.82, 1.115) | 0.5635 | 1.003 (0.8, 1.258) | 0.9759 | 0.972 (0.706, 1.339) | 0.863 | 1.301 (1.035, 1.636) | 0.0242 |
| Chronic Lung Disease | 1.711 (1.298, 2.257) | 0.0001 | 1.393 (0.935, 2.077) | 0.1034 | 1.665 (0.99, 2.801) | 0.0546 | 1.629 (1.073, 2.473) | 0.0219 |
| Diabetes Mellitus | 0.96 (0.792, 1.165) | 0.6821 | 1.22 (0.922, 1.616) | 0.1645 | 0.859 (0.578, 1.277) | 0.4525 | 1.101 (0.814, 1.488) | 0.5336 |
| Dementia | 1.027 (0.678, 1.556) | 0.8994 | 0.801 (0.407, 1.574) | 0.5189 | 0.175 (0.024, 1.271) | 0.0848 | 0.978 (0.578, 1.653) | 0.9328 |
| Psychotic Disorders | 1.267 (0.701, 2.29) | 0.4331 | 1.738 (0.802, 3.765) | 0.1612 | 1.211 (0.367, 3.995) | 0.7531 | 1.503 (0.557, 4.058) | 0.421 |
| Season[2] | | | | | | | | |
| Spring 2020 | 1.334 (1.149, 1.549) | 0.0002 | 1.536 (1.235, 1.91) | 0.0001 | 1.756 (1.31, 2.353) | 0.0002 | 2.608 (2.072, 3.283) | < .0001 |
| Summer 2020 | 0.987 (0.703, 1.387) | 0.9419 | 1.032 (0.612, 1.737) | 0.907 | 0.759 (0.327, 1.76) | 0.5205 | 1.407 (0.829, 2.388) | 0.2063 |
| Fall 2020 | 1.272 (1.058, 1.528) | 0.0104 | 1.255 (0.948, 1.663) | 0.1127 | 1.039 (0.688, 1.568) | 0.8556 | 0.947 (0.657, 1.365) | 0.7709 |
| Spring 2021 | 1.153 (0.804, 1.653) | 0.4387 | 1.082 (0.619, 1.892) | 0.781 | 1.274 (0.628, 2.584) | 0.5022 | 0.639 (0.28, 1.456) | 0.2867 |

c. Patients with Current or Past Aspirin Use

| | Hospitalization | | ICU admission | | Intubation | | Death | |
|---|---|---|---|---|---|---|---|---|
| | OR (95% CI) | P-value | OR (95% CI) | P-value | OR (95% CI) | P-value | OR (95% CI) | P-value |
| Current statin use | 1.052 (0.872, 1.27) | 0.5977 | 0.915 (0.695, 1.205) | 0.5283 | 1.085 (0.73, 1.614) | 0.6855 | 0.619 (0.467, 0.821) | 0.0009 |
| Current RAASi use | 0.906 (0.757, 1.085) | 0.2847 | 0.88 (0.677, 1.144) | 0.339 | 0.869 (0.599, 1.261) | 0.4609 | 0.735 (0.552, 0.979) | 0.0354 |
| Current aspirin use | 1.207 (1.017, 1.433) | 0.0314 | 1.132 (0.882, 1.454) | 0.3304 | 1.004 (0.701, 1.438) | 0.9812 | 1.022 (0.782, 1.336) | 0.8716 |
| Current metformin use | 1.039 (0.807, 1.339) | 0.7645 | 1.052 (0.736, 1.505) | 0.7796 | 1.328 (0.823, 2.143) | 0.2454 | 1.036 (0.681, 1.576) | 0.8682 |
| Age | 1.034 (1.026, 1.042) | < .0001 | 1.021 (1.009, 1.032) | 0.0004 | 1.019 (1.002, 1.036) | 0.0283 | 1.067 (1.053, 1.082) | < .0001 |
| Female | 0.755 (0.635, 0.897) | 0.0014 | 0.618 (0.48, 0.796) | 0.0002 | 0.426 (0.293, 0.62) | < .0001 | 0.489 (0.372, 0.645) | < .0001 |
| English | 0.97 (0.775, 1.215) | 0.7932 | 0.671 (0.487, 0.924) | 0.0145 | 0.926 (0.593, 1.449) | 0.7377 | 1.349 (0.917, 1.983) | 0.1283 |
| White | 0.88 (0.711, 1.088) | 0.2382 | 1.53 (1.116, 2.096) | 0.0082 | 0.844 (0.549, 1.297) | 0.4381 | 1.168 (0.82, 1.663) | 0.3904 |
| Partnered | 0.729 (0.614, 0.865) | 0.0003 | 0.874 (0.68, 1.121) | 0.2888 | 1.17 (0.82, 1.668) | 0.3874 | 0.905 (0.687, 1.192) | 0.4769 |
| Median Household Income By $1000 | 0.992 (0.988, 0.995) | < .0001 | 0.983 (0.977, 0.989) | < .0001 | 0.993 (0.985, 1.001) | 0.071 | 0.999 (0.994, 1.004) | 0.7517 |
| Commercial Insurance | 0.738 (0.618, 0.882) | 0.0008 | 0.727 (0.558, 0.947) | 0.0179 | 0.786 (0.541, 1.142) | 0.2068 | 0.713 (0.528, 0.962) | 0.0271 |
| History of smoking | 1.161 (0.979, 1.375) | 0.0859 | 1.319 (1.026, 1.696) | 0.0311 | 1.339 (0.932, 1.924) | 0.1139 | 1.226 (0.939, 1.601) | 0.1341 |
| HbA1c | 1.085 (1.007, 1.168) | 0.0323 | 1.055 (0.954, 1.168) | 0.2967 | 1.068 (0.935, 1.218) | 0.3324 | 1.202 (1.076, 1.342) | 0.0011 |
| BMI by 10 kg/m$^2$ | 1.192 (1.048, 1.355) | 0.0076 | 1.302 (1.085, 1.561) | 0.0045 | 1.635 (1.284, 2.082) | 0.0001 | 0.984 (0.794, 1.22) | 0.8817 |
| SBP by 10 mm Hg | 0.963 (0.911, 1.018) | 0.1858 | 0.965 (0.89, 1.047) | 0.3946 | 1.022 (0.913, 1.145) | 0.7025 | 0.912 (0.837, 0.993) | 0.0335 |
| DBP by 10 mm Hg | 1.006 (0.909, 1.114) | 0.9049 | 1.076 (0.928, 1.247) | 0.3326 | 0.934 (0.757, 1.152) | 0.5218 | 1.066 (0.907, 1.252) | 0.4386 |
| LDL by 10 mg/dL | 1.009 (0.984, 1.035) | 0.4816 | 1.018 (0.979, 1.059) | 0.3688 | 1.036 (0.982, 1.093) | 0.1947 | 0.983 (0.941, 1.027) | 0.4387 |
| eGFR[1] | 0.734 (0.618, 0.871) | 0.0004 | 0.824 (0.649, 1.048) | 0.1142 | 0.796 (0.574, 1.105) | 0.1732 | 0.631 (0.496, 0.803) | 0.0002 |
| Proteinuria | 1.663 (1.377, 2.007) | < .0001 | 1.527 (1.164, 2.003) | 0.0022 | 1.718 (1.178, 2.505) | 0.0049 | 1.589 (1.208, 2.091) | 0.0009 |
| CCI | 1.054 (1.019, 1.09) | 0.0023 | 1.076 (1.024, 1.13) | 0.0038 | 1.083 (1.008, 1.163) | 0.0286 | 1.144 (1.087, 1.203) | < .0001 |
| ASCVD | 0.977 (0.813, 1.174) | 0.8018 | 1.165 (0.893, 1.52) | 0.2596 | 0.827 (0.565, 1.21) | 0.3273 | 1.242 (0.949, 1.625) | 0.1149 |
| Chronic Lung Disease | 1.593 (1.134, 2.239) | 0.0073 | 1.365 (0.86, 2.164) | 0.1866 | 1.995 (1.112, 3.579) | 0.0206 | 1.571 (0.946, 2.608) | 0.081 |
| Diabetes Mellitus | 1.126 (0.877, 1.445) | 0.354 | 1.308 (0.918, 1.865) | 0.1372 | 1.442 (0.876, 2.374) | 0.1499 | 0.958 (0.659, 1.395) | 0.8247 |
| Dementia | 1.023 (0.644, 1.626) | 0.9228 | 0.8 (0.373, 1.713) | 0.5649 | 0.736 (0.223, 2.428) | 0.6143 | 0.845 (0.464, 1.536) | 0.5805 |
| Psychotic Disorders | 1.135 (0.607, 2.122) | 0.6909 | 1.42 (0.617, 3.268) | 0.4099 | 2.774 (1.111, 6.925) | 0.0288 | 3.654 (1.625, 8.214) | 0.0017 |
| Season[2] | | | | | | | | |
| Spring 2020 | 0.964 (0.797, 1.166) | 0.7048 | 0.95 (0.72, 1.254) | 0.7185 | 1.154 (0.788, 1.69) | 0.4625 | 2.389 (1.808, 3.157) | < .0001 |

*(Continued)*

**Table 2.** (Continued)

| | Hospitalization | | ICU admission | | Intubation | | Death | |
|---|---|---|---|---|---|---|---|---|
| Summer 2020 | 0.444 (0.273, 0.721) | 0.001 | 0.502 (0.239, 1.055) | 0.0691 | 0.557 (0.198, 1.568) | 0.268 | 1.394 (0.736, 2.643) | 0.3083 |
| Fall 2020 | 1.015 (0.792, 1.301) | 0.9064 | 0.991 (0.685, 1.433) | 0.961 | 0.744 (0.418, 1.325) | 0.3159 | 1.231 (0.788, 1.921) | 0.3614 |
| Spring 2021 | 1.176 (0.767, 1.803) | 0.4584 | 1.521 (0.862, 2.685) | 0.1481 | 1.183 (0.496, 2.819) | 0.7051 | 0.664 (0.254, 1.734) | 0.4031 |

d. Patients with Current or Past Metformin Use

| | Hospitalization | | ICU admission | | Intubation | | Death | |
|---|---|---|---|---|---|---|---|---|
| | OR (95% CI) | P-value | OR (95% CI) | P-value | OR (95% CI) | P-value | OR (95% CI) | P-value |
| Current statin use | 0.921 (0.745, 1.139) | 0.4469 | 1.091 (0.802, 1.484) | 0.5788 | 1.172 (0.774, 1.774) | 0.4533 | 1.067 (0.707, 1.61) | 0.756 |
| Current RAASi use | 0.948 (0.779, 1.154) | 0.595 | 1.057 (0.797, 1.401) | 0.7003 | 1.163 (0.797, 1.697) | 0.4338 | 0.6 (0.411, 0.876) | 0.0081 |
| Current aspirin use | 1.313 (1.036, 1.665) | 0.0244 | 0.973 (0.694, 1.365) | 0.8746 | 1.132 (0.724, 1.77) | 0.5864 | 1.42 (0.926, 2.179) | 0.1082 |
| Current metformin use | 1.083 (0.869, 1.348) | 0.4787 | 1.005 (0.735, 1.373) | 0.9764 | 1.218 (0.79, 1.877) | 0.3713 | 0.545 (0.369, 0.805) | 0.0023 |
| Age | 1.037 (1.027, 1.047) | < .0001 | 1.036 (1.022, 1.051) | < .0001 | 1.03 (1.01, 1.05) | 0.0031 | 1.067 (1.046, 1.089) | < .0001 |
| Female | 0.768 (0.634, 0.932) | 0.0073 | 0.702 (0.532, 0.926) | 0.0122 | 0.492 (0.339, 0.714) | 0.0002 | 0.735 (0.503, 1.076) | 0.1132 |
| English | 1.033 (0.812, 1.313) | 0.7936 | 0.822 (0.587, 1.151) | 0.2537 | 0.774 (0.49, 1.221) | 0.2709 | 1.277 (0.776, 2.104) | 0.3359 |
| White | 0.867 (0.693, 1.085) | 0.2132 | 1.004 (0.725, 1.391) | 0.9807 | 0.89 (0.578, 1.37) | 0.5972 | 0.935 (0.591, 1.48) | 0.7749 |
| Partnered | 0.865 (0.718, 1.043) | 0.1298 | 0.894 (0.684, 1.169) | 0.4138 | 1.177 (0.823, 1.684) | 0.3708 | 1.304 (0.9, 1.889) | 0.1611 |
| Median Household Income By $1000 | 0.997 (0.993, 1.002) | 0.1974 | 0.99 (0.983, 0.997) | 0.004 | 1 (0.991, 1.008) | 0.9346 | 1.007 (0.999, 1.015) | 0.0731 |
| Commercial Insurance | 0.802 (0.664, 0.968) | 0.0218 | 0.834 (0.636, 1.094) | 0.1893 | 0.796 (0.555, 1.143) | 0.2172 | 0.66 (0.447, 0.975) | 0.0371 |
| History of smoking | 1.041 (0.86, 1.259) | 0.6807 | 1.346 (1.022, 1.772) | 0.0344 | 1.229 (0.851, 1.774) | 0.2722 | 1.425 (0.978, 2.078) | 0.0654 |
| HbA1c | 1.139 (1.08, 1.202) | < .0001 | 1.087 (1.008, 1.173) | 0.0308 | 1.151 (1.044, 1.269) | 0.0047 | 1.079 (0.966, 1.206) | 0.1789 |
| BMI by 10 kg/m$^2$ | 1.293 (1.131, 1.479) | 0.0002 | 1.261 (1.04, 1.529) | 0.0181 | 1.898 (1.507, 2.389) | < .0001 | 0.949 (0.715, 1.259) | 0.7148 |
| SBP by 10 mm Hg | 1.011 (0.949, 1.078) | 0.731 | 0.999 (0.915, 1.092) | 0.9885 | 0.994 (0.882, 1.121) | 0.9233 | 0.901 (0.799, 1.015) | 0.0854 |
| DBP by 10 mm Hg | 1.037 (0.928, 1.158) | 0.5224 | 1.19 (1.018, 1.39) | 0.0287 | 1.179 (0.959, 1.448) | 0.1176 | 1.186 (0.962, 1.463) | 0.1103 |
| LDL by 10 mg/dL | 1.033 (1.005, 1.061) | 0.0194 | 1.04 (1.002, 1.079) | 0.0399 | 1.054 (1.003, 1.107) | 0.0366 | 1.006 (0.952, 1.062) | 0.8433 |
| eGFR[1] | 0.785 (0.602, 1.024) | 0.0743 | 0.909 (0.635, 1.301) | 0.6026 | 0.958 (0.574, 1.599) | 0.8699 | 0.659 (0.442, 0.983) | 0.041 |
| Proteinuria | 1.697 (1.384, 2.081) | < .0001 | 2.374 (1.806, 3.121) | < .0001 | 2.399 (1.674, 3.436) | < .0001 | 1.961 (1.361, 2.826) | 0.0003 |
| CCI | 1.067 (1.025, 1.112) | 0.0017 | 1.086 (1.026, 1.149) | 0.0044 | 1.129 (1.045, 1.219) | 0.0021 | 1.158 (1.079, 1.243) | 0.0001 |
| ASCVD | 0.951 (0.755, 1.199) | 0.6704 | 1.065 (0.775, 1.464) | 0.6965 | 0.986 (0.641, 1.517) | 0.9498 | 0.885 (0.599, 1.307) | 0.538 |
| Chronic Lung Disease | 1.198 (0.772, 1.861) | 0.4202 | 1.059 (0.571, 1.965) | 0.8559 | 1.804 (0.907, 3.586) | 0.0925 | 1.34 (0.608, 2.954) | 0.4678 |
| Diabetes Mellitus | 0.842 (0.608, 1.166) | 0.3008 | 0.867 (0.535, 1.405) | 0.562 | 0.634 (0.345, 1.165) | 0.1422 | 1.612 (0.689, 3.773) | 0.2707 |
| Dementia | 0.491 (0.221, 1.091) | 0.0808 | 0.577 (0.198, 1.683) | 0.3138 | 0 (0,.) | 0.9836 | 1.257 (0.496, 3.19) | 0.6297 |
| Psychotic Disorders | 1.762 (0.955, 3.254) | 0.0701 | 1.693 (0.724, 3.961) | 0.2244 | 1.31 (0.388, 4.425) | 0.664 | 3.549 (1.362, 9.243) | 0.0095 |
| Season[2] | | | | | | | | |
| Spring 2020 | 1.524 (1.238, 1.877) | 0.0001 | 1.813 (1.352, 2.432) | 0.0001 | 1.952 (1.34, 2.844) | 0.0005 | 2.81 (1.903, 4.149) | < .0001 |
| Summer 2020 | 0.838 (0.503, 1.397) | 0.4979 | 0.803 (0.358, 1.797) | 0.5928 | 0.619 (0.188, 2.035) | 0.4292 | 1.512 (0.587, 3.894) | 0.392 |
| Fall 2020 | 1.155 (0.883, 1.511) | 0.2929 | 1.383 (0.942, 2.029) | 0.0975 | 0.792 (0.44, 1.424) | 0.4359 | 1.112 (0.612, 2.023) | 0.7267 |
| Spring 2021 | 1.017 (0.6, 1.723) | 0.95 | 1.268 (0.617, 2.606) | 0.5176 | 0.987 (0.374, 2.604) | 0.9789 | 1.344 (0.472, 3.827) | 0.5799 |

[1]eGFR was log transformed to reduce data skewness

[2]Winter 2020–21 served as the reference

Abbreviations: ASCVD, atherosclerotic cardiovascular disease; BMI, body mass index; CCI, Charlson comorbidity index; DBP, diastolic blood pressure; eGFR, estimated glomerular filtration rate; HbA1c, hemoglobin A1c; LDL, low density lipoprotein cholesterol; RAASi, renin angiotensin aldosterone system inhibitor; SBP, systolic blood pressure

infection than those who were not [25]. De Spiegeleer *et al.* suggested that the discrepancy they saw between an association with asymptomatic disease and not mortality was due to the statins' potential therapeutic effects during the initial stages of COVID-19 infection and not later in the disease progression [25]. RAASi use may have been associated with decreased risk of mortality secondary to a reduction in acute lung injury [37]. Metformin's cardioprotective

**Table 3. Effect of study medications on COVID-19 outcomes: Propensity score analyses.**

|  | Statins | | RAASi | | Aspirin | | Metformin | |
|---|---|---|---|---|---|---|---|---|
|  | OR (95% CI) | P-value | OR (95% CI) | P-value | OR (95% CI) | P-value | OR (95% CI) | P-value |
| **Hospitalization** | 0.916 (0.791, 1.06) | 0.2401 | 0.984 (0.857, 1.129) | 0.8141 | 1.183 (1.004, 1.393) | 0.0446 | 1.039 (0.844, 1.279) | 0.7189 |
| **ICU admission** | 0.892 (0.719, 1.105) | 0.2941 | 0.946 (0.767, 1.165) | 0.5992 | 1.122 (0.879, 1.432) | 0.3567 | 0.972 (0.722, 1.307) | 0.849 |
| **Intubation** | 0.909 (0.669, 1.235) | 0.5407 | 1.084 (0.804, 1.463) | 0.5963 | 1.009 (0.709, 1.434) | 0.9614 | 1.157 (0.762, 1.756) | 0.4939 |
| **Death** | 0.638 (0.522, 0.78) | < .0001 | 0.564 (0.459, 0.693) | < .0001 | 0.923 (0.726, 1.175) | 0.5156 | 0.589 (0.414, 0.838) | 0.0032 |

effects [13] may have been the reason for the observed lower risk of death, but did not have an impact on hospitalization-related COVID-19 outcomes.

Khunti *et al.* observed an improvement in COVID-19 outcomes among patients taking metformin and a greater risk of adverse outcomes associated with insulin use in a nationwide cohort of individuals with T2DM [21]. They proposed that these associations were not actually due to the medications themselves but instead more severe COVID-19 was seen in patients who had progressed from metformin to insulin use to manage their diabetes [21]. It is possible that our findings are also confounded by the differences in disease severity between patients currently vs. previously taking statins, RAASi, and/or metformin. In three out of four medication classes we studied–except aspirin–patients previously on therapy had higher comorbidity load (as represented by CCI) compared to the patients currently on therapy. Other comorbidities not captured by the CCI could therefore have been the reason for the higher mortality observed in this group. Despite access to a robust EMR data source for gathering data on confounders and controlling for a variety of known risk factors for poor COVID-19 outcomes, it is possible that residual confounding remained. For example, patients who continued cardioprotective medications may have been more adherent to protective lifestyle behaviors and other beneficial medications, resulting in lower mortality.

While this study did not find consistent evidence of benefit of cardiovascular medications in treatment of COVID-19, it is possible that other pharmacological approaches focusing on some of the same targets could succeed. In particular, recent studies on DNA and RNA aptamers targeting angiotensin converting enzyme and SARS-COV-2 spike protein have shown promise in *in silico* and *in vitro* studies [38, 39].

This study has many strengths. It was able to reduce selection bias by comparing current to past recipients of the four medication classes under investigation. Our comparison groups allowed for the inclusion of a more general population than other studies that limited the study of RAASi medications to individuals with hypertension [22, 23, 31, 32] and metformin to individuals with T2DM [4, 14, 20]. We instead included all patients whose healthcare providers felt they had an indication for a study medication. Additionally, we were able to minimize exposure misclassification by our definition of past and current medication users. This is an improvement over some earlier studies that only examined medication use at the time of or after hospitalization due to COVID-19 occurred [19, 22–24, 29], rather than at the time of COVID-19 testing. This reduces the risk that any benefits or harms identified to be associated with study medications were due to treatment decisions associated with worsening disease states. This large study was also the first, to our knowledge, to examine a broad range of COVID-19 outcomes and consider four different medication classes in the same source population.

Findings of the study should be interpreted in the light of its limitations. All patients came from a single healthcare delivery system, which could limit our ability to generalize to the entire US or world population. However, study findings align with those in previous studies and a single center source allowed for similar treatment standards among all study patients.

**Table 4. Effect of study medications and patient characteristics on COVID-19 outcomes: Analysis limited to patients with history of adverse reactions to study medications.**

a. Patients with Current or Past Statin Use

| | Hospitalization | | ICU admission | | Intubation | | Death | |
|---|---|---|---|---|---|---|---|---|
| | OR (95% CI) | P-value | OR (95% CI) | P-value | OR (95% CI) | P-value | OR (95% CI) | P-value |
| Current statin use | 0.92 (0.708, 1.196) | 0.5347 | 0.786 (0.551, 1.121) | 0.1835 | 0.864 (0.509, 1.467) | 0.5874 | 0.797 (0.536, 1.187) | 0.2647 |
| Current RAASi use | 0.95 (0.828, 1.09) | 0.4645 | 0.968 (0.792, 1.184) | 0.7552 | 0.962 (0.728, 1.27) | 0.7838 | 0.792 (0.625, 1.004) | 0.0541 |
| Current aspirin use | 1.265 (1.067, 1.5) | 0.0067 | 1.16 (0.909, 1.479) | 0.2323 | 1.038 (0.738, 1.461) | 0.8292 | 1.183 (0.896, 1.562) | 0.2359 |
| Current metformin use | 1.176 (0.964, 1.435) | 0.1095 | 1.173 (0.89, 1.546) | 0.2566 | 1.3 (0.894, 1.89) | 0.1691 | 0.69 (0.486, 0.978) | 0.0373 |
| Age | 1.035 (1.027, 1.042) | < .0001 | 1.019 (1.009, 1.03) | 0.0004 | 1.015 (1, 1.03) | 0.0484 | 1.078 (1.064, 1.092) | < .0001 |
| Female | 0.797 (0.693, 0.916) | 0.0015 | 0.772 (0.629, 0.948) | 0.0133 | 0.587 (0.439, 0.784) | 0.0003 | 0.605 (0.475, 0.772) | 0.0001 |
| English | 0.926 (0.761, 1.126) | 0.4413 | 0.715 (0.546, 0.937) | 0.015 | 0.802 (0.557, 1.155) | 0.2362 | 1.125 (0.783, 1.617) | 0.5233 |
| White | 0.773 (0.647, 0.925) | 0.0049 | 1.147 (0.885, 1.487) | 0.3006 | 0.701 (0.497, 0.987) | 0.042 | 1.244 (0.888, 1.745) | 0.2045 |
| Partnered | 0.817 (0.712, 0.937) | 0.0039 | 0.909 (0.744, 1.111) | 0.3523 | 1.228 (0.927, 1.628) | 0.1528 | 0.941 (0.742, 1.194) | 0.6181 |
| Median Household Income By $1000 | 0.995 (0.992, 0.998) | 0.0017 | 0.986 (0.981, 0.991) | < .0001 | 0.995 (0.988, 1.001) | 0.0948 | 0.997 (0.993, 1.002) | 0.2331 |
| Commercial Insurance | 0.774 (0.67, 0.894) | 0.0005 | 0.643 (0.519, 0.797) | 0.0001 | 0.622 (0.463, 0.837) | 0.0017 | 0.562 (0.425, 0.742) | 0.0001 |
| History of smoking | 1.057 (0.922, 1.212) | 0.4251 | 1.155 (0.945, 1.413) | 0.1587 | 1.059 (0.802, 1.399) | 0.686 | 1.173 (0.925, 1.489) | 0.188 |
| HbA1c | 1.111 (1.046, 1.18) | 0.0007 | 1.035 (0.959, 1.118) | 0.3773 | 1.084 (0.979, 1.201) | 0.1195 | 1.185 (1.07, 1.313) | 0.0012 |
| BMI by 10 kg/m$^2$ | 1.223 (1.097, 1.363) | 0.0003 | 1.246 (1.07, 1.451) | 0.0047 | 1.845 (1.527, 2.231) | < .0001 | 0.905 (0.736, 1.113) | 0.3437 |
| SBP by 10 mm Hg | 0.981 (0.938, 1.027) | 0.4139 | 0.971 (0.909, 1.037) | 0.3732 | 1.044 (0.955, 1.142) | 0.3454 | 0.893 (0.828, 0.963) | 0.0033 |
| DBP by 10 mm Hg | 1.014 (0.934, 1.101) | 0.7437 | 1.067 (0.946, 1.203) | 0.2918 | 0.914 (0.775, 1.078) | 0.2865 | 1.03 (0.895, 1.186) | 0.6791 |
| LDL by 10 mg/dL | 1.022 (1.001, 1.043) | 0.04 | 1.045 (1.015, 1.076) | 0.0027 | 1.034 (0.993, 1.077) | 0.1079 | 0.988 (0.95, 1.028) | 0.5456 |
| eGFR[1] | 0.627 (0.535, 0.734) | < .0001 | 0.73 (0.589, 0.905) | 0.0041 | 0.853 (0.632, 1.15) | 0.2973 | 0.628 (0.497, 0.793) | 0.0001 |
| Proteinuria | 1.727 (1.476, 2.019) | < .0001 | 1.788 (1.434, 2.229) | < .0001 | 2.005 (1.488, 2.702) | < .0001 | 1.422 (1.107, 1.827) | 0.0059 |
| CCI | 1.067 (1.039, 1.097) | < .0001 | 1.068 (1.027, 1.111) | 0.0011 | 1.081 (1.022, 1.143) | 0.0068 | 1.123 (1.076, 1.172) | < .0001 |
| ASCVD | 1.024 (0.879, 1.193) | 0.7572 | 1.062 (0.849, 1.328) | 0.5971 | 1.028 (0.752, 1.405) | 0.8629 | 1.295 (1.016, 1.651) | 0.0366 |
| Chronic Lung Disease | 1.7 (1.293, 2.237) | 0.0001 | 1.349 (0.906, 2.008) | 0.1404 | 1.88 (1.14, 3.099) | 0.0134 | 1.55 (1.002, 2.397) | 0.0489 |
| Diabetes Mellitus | 0.966 (0.786, 1.187) | 0.7403 | 1.315 (0.988, 1.751) | 0.0609 | 1.042 (0.698, 1.554) | 0.8421 | 1.438 (1.041, 1.985) | 0.0273 |
| Dementia | 1.228 (0.81, 1.863) | 0.3329 | 1.159 (0.631, 2.128) | 0.6349 | 0.538 (0.167, 1.739) | 0.3007 | 1.173 (0.677, 2.032) | 0.5698 |
| Psychotic Disorders | 1.856 (1.107, 3.112) | 0.0191 | 1.672 (0.85, 3.287) | 0.1362 | 1.544 (0.596, 4.003) | 0.3712 | 2.731 (1.235, 6.037) | 0.0131 |
| Season[2] | | | | | | | | |
| Spring 2020 | 1.367 (1.17, 1.598) | 0.0001 | 1.517 (1.214, 1.895) | 0.0002 | 1.743 (1.296, 2.345) | 0.0002 | 2.992 (2.334, 3.835) | < .0001 |
| Summer 2020 | 0.876 (0.614, 1.249) | 0.4639 | 0.908 (0.531, 1.55) | 0.7229 | 0.708 (0.305, 1.645) | 0.4221 | 1.211 (0.666, 2.201) | 0.5309 |
| Fall 2020 | 1.128 (0.927, 1.372) | 0.2284 | 1.201 (0.9, 1.601) | 0.2133 | 0.848 (0.542, 1.326) | 0.469 | 1.137 (0.775, 1.667) | 0.5112 |
| Spring 2021 | 1.053 (0.716, 1.549) | 0.7925 | 1.323 (0.777, 2.251) | 0.3021 | 1.01 (0.457, 2.232) | 0.9802 | 0.862 (0.382, 1.945) | 0.7199 |

b. Patients with Current or Past RAASi use

| | Hospitalization | | ICU admission | | Intubation | | Death | |
|---|---|---|---|---|---|---|---|---|
| | OR (95% CI) | P-value | OR (95% CI) | P-value | OR (95% CI) | P-value | OR (95% CI) | P-value |
| Current statin use | 0.825 (0.702, 0.968) | 0.0186 | 1.002 (0.786, 1.277) | 0.9872 | 0.966 (0.696, 1.342) | 0.8387 | 0.791 (0.592, 1.057) | 0.1129 |
| Current RAASi use | 1.069 (0.873, 1.31) | 0.5166 | 0.983 (0.728, 1.327) | 0.9106 | 1.062 (0.693, 1.628) | 0.7826 | 0.862 (0.63, 1.18) | 0.3543 |
| Current aspirin use | 1.182 (0.968, 1.443) | 0.1012 | 1.007 (0.748, 1.354) | 0.9656 | 0.923 (0.612, 1.391) | 0.7008 | 1.156 (0.815, 1.638) | 0.4161 |
| Current metformin use | 1.213 (0.983, 1.497) | 0.0724 | 1.282 (0.949, 1.731) | 0.106 | 1.385 (0.93, 2.064) | 0.1093 | 0.684 (0.458, 1.022) | 0.0636 |
| Age | 1.038 (1.03, 1.045) | < .0001 | 1.033 (1.022, 1.045) | < .0001 | 1.018 (1.003, 1.034) | 0.0208 | 1.071 (1.056, 1.087) | < .0001 |
| Female | 0.807 (0.695, 0.938) | 0.0053 | 0.666 (0.53, 0.836) | 0.0005 | 0.597 (0.438, 0.816) | 0.0012 | 0.616 (0.466, 0.813) | 0.0006 |
| English | 0.982 (0.801, 1.205) | 0.8641 | 0.847 (0.632, 1.135) | 0.2652 | 1.02 (0.684, 1.522) | 0.923 | 1.329 (0.889, 1.987) | 0.1657 |
| White | 0.8 (0.665, 0.962) | 0.0177 | 1.019 (0.774, 1.342) | 0.8925 | 0.728 (0.508, 1.044) | 0.0844 | 1.186 (0.827, 1.703) | 0.354 |
| Partnered | 0.876 (0.757, 1.015) | 0.0777 | 0.986 (0.791, 1.23) | 0.9032 | 1.241 (0.917, 1.678) | 0.1617 | 0.868 (0.663, 1.137) | 0.3056 |
| Median Household Income By $1000 | 0.995 (0.992, 0.998) | 0.0015 | 0.987 (0.982, 0.992) | < .0001 | 0.993 (0.986, 1) | 0.0421 | 0.997 (0.991, 1.002) | 0.2052 |
| Commercial Insurance | 0.776 (0.666, 0.904) | 0.0011 | 0.828 (0.659, 1.041) | 0.1061 | 0.802 (0.589, 1.092) | 0.162 | 0.642 (0.473, 0.871) | 0.0043 |

*(Continued)*

**Table 4.** (Continued)

| | Hospitalization OR (95% CI) | P-value | ICU admission OR (95% CI) | P-value | Intubation OR (95% CI) | P-value | Death OR (95% CI) | P-value |
|---|---|---|---|---|---|---|---|---|
| History of smoking | 1.031 (0.89, 1.193) | 0.685 | 1.066 (0.854, 1.33) | 0.5726 | 0.965 (0.714, 1.306) | 0.8182 | 1.186 (0.908, 1.55) | 0.2114 |
| HbA1c | 1.131 (1.064, 1.202) | 0.0001 | 1.09 (1.004, 1.183) | 0.0398 | 1.164 (1.052, 1.288) | 0.0034 | 1.177 (1.046, 1.323) | 0.0069 |
| BMI by 10 kg/m$^2$ | 1.325 (1.19, 1.476) | < .0001 | 1.326 (1.129, 1.557) | 0.0006 | 1.859 (1.53, 2.259) | < .0001 | 0.987 (0.793, 1.229) | 0.9102 |
| SBP by 10 mm Hg | 1.016 (0.97, 1.065) | 0.4973 | 0.979 (0.914, 1.05) | 0.5574 | 1.063 (0.969, 1.165) | 0.1939 | 0.922 (0.848, 1.002) | 0.0561 |
| DBP by 10 mm Hg | 1.039 (0.954, 1.132) | 0.3776 | 1.212 (1.066, 1.377) | 0.0032 | 1.05 (0.886, 1.245) | 0.5741 | 0.928 (0.795, 1.083) | 0.3421 |
| LDL by 10 mg/dL | 1.008 (0.986, 1.031) | 0.4702 | 1.04 (1.007, 1.074) | 0.0185 | 1.042 (0.997, 1.089) | 0.0653 | 1.008 (0.965, 1.053) | 0.7204 |
| eGFR[1] | 0.657 (0.551, 0.784) | < .0001 | 0.686 (0.535, 0.878) | 0.0028 | 0.791 (0.566, 1.105) | 0.1688 | 0.514 (0.395, 0.67) | < .0001 |
| Proteinuria | 2.094 (1.782, 2.462) | < .0001 | 1.789 (1.412, 2.268) | < .0001 | 2.185 (1.601, 2.981) | < .0001 | 1.593 (1.203, 2.109) | 0.0011 |
| CCI | 1.059 (1.028, 1.091) | 0.0001 | 1.086 (1.039, 1.136) | 0.0003 | 1.126 (1.057, 1.2) | 0.0002 | 1.097 (1.044, 1.152) | 0.0003 |
| ASCVD | 0.962 (0.809, 1.144) | 0.6608 | 0.927 (0.716, 1.201) | 0.5675 | 0.973 (0.68, 1.392) | 0.8804 | 1.098 (0.829, 1.455) | 0.5149 |
| Chronic Lung Disease | 1.698 (1.234, 2.337) | 0.0012 | 1.74 (1.128, 2.682) | 0.0122 | 1.88 (1.072, 3.296) | 0.0275 | 2.164 (1.333, 3.512) | 0.0018 |
| Diabetes Mellitus | 0.961 (0.772, 1.196) | 0.7216 | 1.289 (0.939, 1.77) | 0.1162 | 1.059 (0.684, 1.639) | 0.7987 | 1.216 (0.842, 1.758) | 0.2976 |
| Dementia | 0.91 (0.54, 1.531) | 0.7213 | 0.6 (0.236, 1.53) | 0.285 | 0 (0,.) | 0.979 | 0.997 (0.489, 2.032) | 0.9927 |
| Psychotic Disorders | 1.599 (0.821, 3.113) | 0.1672 | 2.121 (0.914, 4.921) | 0.0799 | 1.633 (0.485, 5.495) | 0.4284 | 2.541 (0.859, 7.514) | 0.0919 |
| Season[2] | | | | | | | | |
| Spring 2020 | 1.429 (1.209, 1.689) | < .0001 | 1.812 (1.422, 2.311) | < .0001 | 2.154 (1.562, 2.97) | < .0001 | 2.858 (2.152, 3.796) | < .0001 |
| Summer 2020 | 0.969 (0.654, 1.436) | 0.8754 | 1.012 (0.547, 1.872) | 0.9704 | 0.846 (0.335, 2.135) | 0.7232 | 2.028 (1.088, 3.781) | 0.0261 |
| Fall 2020 | 1.311 (1.071, 1.604) | 0.0087 | 1.333 (0.977, 1.817) | 0.0696 | 1.058 (0.667, 1.676) | 0.8116 | 1.174 (0.773, 1.784) | 0.452 |
| Spring 2021 | 1.225 (0.83, 1.809) | 0.307 | 1.176 (0.644, 2.149) | 0.5975 | 1.619 (0.786, 3.336) | 0.1916 | 0.884 (0.367, 2.132) | 0.7837 |

c. Patients with Current or Past Aspirin Use

| | Hospitalization OR (95% CI) | P-value | ICU admission OR (95% CI) | P-value | Intubation OR (95% CI) | P-value | Death OR (95% CI) | P-value |
|---|---|---|---|---|---|---|---|---|
| Current statin use | 0.927 (0.716, 1.201) | 0.5671 | 0.9 (0.619, 1.31) | 0.583 | 0.979 (0.567, 1.692) | 0.9403 | 0.643 (0.429, 0.963) | 0.0321 |
| Current RAASi use | 0.866 (0.682, 1.099) | 0.2358 | 0.814 (0.576, 1.15) | 0.243 | 0.821 (0.499, 1.352) | 0.4391 | 0.735 (0.495, 1.092) | 0.1273 |
| Current aspirin use | 1.304 (0.987, 1.723) | 0.0613 | 1.185 (0.785, 1.788) | 0.4188 | 0.986 (0.54, 1.8) | 0.9635 | 1.229 (0.779, 1.938) | 0.3757 |
| Current metformin use | 1.055 (0.76, 1.464) | 0.7505 | 1.01 (0.638, 1.6) | 0.9646 | 1.308 (0.701, 2.44) | 0.3983 | 1.149 (0.652, 2.024) | 0.6306 |
| Age | 1.03 (1.02, 1.041) | < .0001 | 1.014 (0.999, 1.029) | 0.0719 | 1.022 (0.999, 1.044) | 0.0582 | 1.082 (1.061, 1.102) | < .0001 |
| Female | 0.703 (0.556, 0.889) | 0.0032 | 0.704 (0.503, 0.986) | 0.0413 | 0.43 (0.26, 0.712) | 0.001 | 0.666 (0.453, 0.98) | 0.039 |
| English | 0.944 (0.709, 1.256) | 0.6929 | 0.698 (0.462, 1.056) | 0.089 | 1.104 (0.616, 1.98) | 0.7396 | 1.068 (0.645, 1.769) | 0.7971 |
| White | 0.973 (0.735, 1.287) | 0.8474 | 1.863 (1.235, 2.81) | 0.003 | 0.791 (0.444, 1.407) | 0.4244 | 1.732 (1.052, 2.85) | 0.0308 |
| Partnered | 0.733 (0.581, 0.924) | 0.0087 | 0.924 (0.662, 1.288) | 0.6394 | 1.37 (0.852, 2.204) | 0.1938 | 0.785 (0.529, 1.163) | 0.2267 |
| Median Household Income By $1000 | 0.987 (0.982, 0.992) | < .0001 | 0.984 (0.976, 0.992) | 0.0001 | 0.991 (0.981, 1.003) | 0.1299 | 0.993 (0.986, 1.001) | 0.0675 |
| Commercial Insurance | 0.746 (0.589, 0.946) | 0.0153 | 0.65 (0.456, 0.925) | 0.0167 | 0.581 (0.345, 0.978) | 0.0408 | 0.594 (0.388, 0.91) | 0.0167 |
| History of smoking | 1.157 (0.922, 1.452) | 0.208 | 1.248 (0.896, 1.737) | 0.1895 | 1.12 (0.694, 1.807) | 0.6418 | 1.011 (0.699, 1.463) | 0.9525 |
| HbA1c | 1.056 (0.965, 1.155) | 0.2355 | 1.011 (0.89, 1.149) | 0.8686 | 0.997 (0.84, 1.182) | 0.9694 | 1.039 (0.875, 1.233) | 0.6612 |
| BMI by 10 kg/m$^2$ | 1.251 (1.059, 1.478) | 0.0085 | 1.222 (0.967, 1.543) | 0.0932 | 1.772 (1.304, 2.406) | 0.0003 | 0.942 (0.699, 1.271) | 0.6973 |
| SBP by 10 mm Hg | 0.94 (0.871, 1.014) | 0.1109 | 0.958 (0.857, 1.07) | 0.442 | 0.973 (0.832, 1.137) | 0.7275 | 0.89 (0.789, 1.003) | 0.0568 |
| DBP by 10 mm Hg | 1.067 (0.931, 1.223) | 0.3508 | 1.198 (0.985, 1.457) | 0.0702 | 0.937 (0.702, 1.25) | 0.6579 | 0.98 (0.783, 1.227) | 0.863 |
| LDL by 10 mg/dL | 1.016 (0.982, 1.052) | 0.3553 | 1.033 (0.982, 1.086) | 0.2054 | 1.054 (0.982, 1.131) | 0.1427 | 0.987 (0.93, 1.047) | 0.6603 |
| eGFR[1] | 0.714 (0.564, 0.903) | 0.005 | 0.727 (0.53, 0.997) | 0.048 | 0.663 (0.431, 1.02) | 0.0615 | 0.644 (0.454, 0.912) | 0.0133 |
| Proteinuria | 1.613 (1.249, 2.085) | 0.0003 | 1.378 (0.949, 2.001) | 0.092 | 1.452 (0.856, 2.466) | 0.1669 | 1.641 (1.111, 2.424) | 0.0128 |
| CCI | 1.049 (1.003, 1.098) | 0.0367 | 1.044 (0.977, 1.116) | 0.1991 | 1.005 (0.913, 1.107) | 0.9185 | 1.059 (0.99, 1.131) | 0.0932 |
| ASCVD | 0.987 (0.772, 1.262) | 0.9159 | 1.252 (0.878, 1.785) | 0.2152 | 1.119 (0.675, 1.856) | 0.6634 | 1.472 (1.003, 2.16) | 0.048 |
| Chronic Lung Disease | 1.877 (1.223, 2.881) | 0.004 | 1.596 (0.901, 2.825) | 0.1088 | 2.181 (1.036, 4.592) | 0.04 | 1.157 (0.547, 2.444) | 0.703 |
| Diabetes Mellitus | 1.206 (0.867, 1.679) | 0.2663 | 1.718 (1.077, 2.742) | 0.0232 | 1.985 (1.019, 3.866) | 0.0437 | 1.156 (0.676, 1.978) | 0.5961 |
| Dementia | 1.235 (0.669, 2.279) | 0.5003 | 0.786 (0.271, 2.284) | 0.6587 | 0.495 (0.065, 3.755) | 0.4964 | 0.237 (0.068, 0.822) | 0.0232 |
| Psychotic Disorders | 0.969 (0.432, 2.169) | 0.9381 | 0.894 (0.259, 3.084) | 0.8588 | 1.85 (0.512, 6.683) | 0.3476 | 2.409 (0.723, 8.026) | 0.1522 |
| Season[2] | | | | | | | | |

*(Continued)*

**Table 4.** (Continued)

| | Hospitalization | | ICU admission | | Intubation | | Death | |
|---|---|---|---|---|---|---|---|---|
| Spring 2020 | 1.059 (0.826, 1.358) | 0.6519 | 0.917 (0.635, 1.323) | 0.6415 | 1.119 (0.679, 1.846) | 0.6584 | 2.069 (1.398, 3.064) | 0.0003 |
| Summer 2020 | 0.637 (0.357, 1.137) | 0.1273 | 0.663 (0.276, 1.593) | 0.3579 | 0.88 (0.295, 2.622) | 0.8184 | 2.256 (1.035, 4.916) | 0.0407 |
| Fall 2020 | 0.975 (0.695, 1.367) | 0.8835 | 1.063 (0.66, 1.71) | 0.8021 | 0.598 (0.26, 1.376) | 0.2264 | 1.351 (0.731, 2.497) | 0.3373 |
| Spring 2021 | 1.01 (0.557, 1.834) | 0.9732 | 1.148 (0.499, 2.641) | 0.7463 | 0.355 (0.047, 2.652) | 0.3128 | 0.523 (0.116, 2.358) | 0.3989 |

d. Patients with Current or Past Metformin Use

| | Hospitalization | | ICU admission | | Intubation | | Death | |
|---|---|---|---|---|---|---|---|---|
| | OR (95% CI) | P-value | OR (95% CI) | P-value | OR (95% CI) | P-value | OR (95% CI) | P-value |
| Current statin use | 0.802 (0.631, 1.02) | 0.0716 | 0.941 (0.668, 1.326) | 0.7295 | 1.059 (0.674, 1.665) | 0.8028 | 0.982 (0.571, 1.689) | 0.9475 |
| Current RAASi use | 1.013 (0.813, 1.263) | 0.9061 | 1.076 (0.787, 1.473) | 0.6449 | 1.337 (0.88, 2.031) | 0.1739 | 0.652 (0.412, 1.031) | 0.0675 |
| Current aspirin use | 1.38 (1.061, 1.795) | 0.0162 | 1.03 (0.708, 1.499) | 0.8784 | 1.211 (0.747, 1.963) | 0.4377 | 1.656 (0.989, 2.772) | 0.0552 |
| Current metformin use | 0.799 (0.55, 1.16) | 0.2382 | 0.597 (0.369, 0.967) | 0.0361 | 0.661 (0.345, 1.267) | 0.2124 | 0.577 (0.296, 1.121) | 0.1047 |
| Age | 1.042 (1.03, 1.054) | < .0001 | 1.039 (1.022, 1.056) | < .0001 | 1.035 (1.012, 1.057) | 0.0024 | 1.078 (1.049, 1.108) | < .0001 |
| Female | 0.749 (0.603, 0.93) | 0.0089 | 0.693 (0.508, 0.945) | 0.0204 | 0.508 (0.338, 0.764) | 0.0011 | 0.657 (0.406, 1.063) | 0.0869 |
| English | 1.001 (0.765, 1.31) | 0.9935 | 0.853 (0.588, 1.239) | 0.4049 | 0.778 (0.473, 1.279) | 0.322 | 0.981 (0.532, 1.808) | 0.9514 |
| White | 0.895 (0.695, 1.153) | 0.3908 | 0.925 (0.642, 1.332) | 0.6738 | 0.768 (0.478, 1.236) | 0.277 | 0.993 (0.549, 1.795) | 0.9814 |
| Partnered | 0.879 (0.712, 1.085) | 0.23 | 0.852 (0.631, 1.15) | 0.2954 | 1.207 (0.814, 1.79) | 0.35 | 1.282 (0.804, 2.043) | 0.2974 |
| Median Household Income By $1000 | 0.996 (0.991, 1.001) | 0.1125 | 0.99 (0.983, 0.998) | 0.0119 | 1.003 (0.994, 1.012) | 0.5758 | 1.012 (1.002, 1.022) | 0.0137 |
| Commercial Insurance | 0.878 (0.711, 1.085) | 0.2297 | 0.942 (0.697, 1.274) | 0.7 | 0.867 (0.583, 1.289) | 0.4818 | 0.573 (0.345, 0.951) | 0.0312 |
| History of smoking | 1.031 (0.832, 1.278) | 0.7771 | 1.308 (0.961, 1.779) | 0.0881 | 1.25 (0.834, 1.873) | 0.2803 | 1.285 (0.801, 2.062) | 0.2983 |
| HbA1c | 1.131 (1.063, 1.203) | 0.0001 | 1.082 (0.992, 1.181) | 0.0759 | 1.175 (1.055, 1.309) | 0.0033 | 1.234 (1.074, 1.419) | 0.0031 |
| BMI by 10 kg/m$^2$ | 1.295 (1.112, 1.508) | 0.0009 | 1.239 (0.999, 1.536) | 0.0515 | 1.927 (1.499, 2.478) | < .0001 | 1.385 (0.992, 1.935) | 0.0557 |
| SBP by 10 mm Hg | 1.002 (0.932, 1.078) | 0.954 | 0.96 (0.867, 1.064) | 0.4353 | 0.967 (0.845, 1.106) | 0.6228 | 0.854 (0.727, 1.002) | 0.0531 |
| DBP by 10 mm Hg | 1.041 (0.916, 1.184) | 0.5359 | 1.166 (0.973, 1.397) | 0.0969 | 1.198 (0.948, 1.512) | 0.1296 | 1.322 (1.004, 1.74) | 0.0466 |
| LDL by 10 mg/dL | 1.054 (1.022, 1.087) | 0.0007 | 1.065 (1.022, 1.111) | 0.003 | 1.076 (1.02, 1.135) | 0.007 | 0.984 (0.912, 1.062) | 0.682 |
| eGFR[1] | 0.794 (0.569, 1.108) | 0.1741 | 0.962 (0.619, 1.497) | 0.8653 | 1.18 (0.637, 2.188) | 0.5989 | 0.7 (0.394, 1.246) | 0.2254 |
| Proteinuria | 1.725 (1.372, 2.169) | < .0001 | 2.418 (1.785, 3.277) | < .0001 | 2.318 (1.568, 3.429) | < .0001 | 1.924 (1.228, 3.013) | 0.0043 |
| CCI | 1.041 (0.994, 1.09) | 0.091 | 1.058 (0.992, 1.128) | 0.0886 | 1.146 (1.051, 1.248) | 0.0019 | 1.187 (1.086, 1.297) | 0.0002 |
| ASCVD | 1.018 (0.784, 1.322) | 0.8942 | 1.146 (0.801, 1.641) | 0.4563 | 1.101 (0.688, 1.761) | 0.689 | 0.955 (0.591, 1.544) | 0.8501 |
| Chronic Lung Disease | 1.228 (0.75, 2.011) | 0.4152 | 1.02 (0.502, 2.071) | 0.9559 | 1.732 (0.808, 3.711) | 0.158 | 1.316 (0.495, 3.496) | 0.5819 |
| Diabetes Mellitus | 0.815 (0.557, 1.193) | 0.2926 | 0.868 (0.497, 1.515) | 0.6188 | 0.573 (0.289, 1.135) | 0.1102 | 1.062 (0.347, 3.25) | 0.9159 |
| Dementia | 0.674 (0.261, 1.737) | 0.4139 | 0.798 (0.226, 2.818) | 0.7262 | 0 (0,.) | 0.9782 | 0.363 (0.043, 3.073) | 0.3522 |
| Psychotic Disorders | 1.98 (0.958, 4.092) | 0.0653 | 1.974 (0.774, 5.037) | 0.1548 | 1.733 (0.497, 6.049) | 0.3883 | 3.921 (1.206, 12.747) | 0.0231 |
| Season[2] | | | | | | | | |
| Spring 2020 | 1.462 (1.155, 1.852) | 0.0016 | 1.916 (1.378, 2.663) | 0.0001 | 2.09 (1.382, 3.16) | 0.0005 | 2.839 (1.742, 4.627) | < .0001 |
| Summer 2020 | 0.826 (0.474, 1.441) | 0.5017 | 0.967 (0.425, 2.197) | 0.9355 | 0.698 (0.209, 2.334) | 0.5599 | 2.133 (0.795, 5.724) | 0.1324 |
| Fall 2020 | 1.218 (0.905, 1.638) | 0.193 | 1.49 (0.974, 2.279) | 0.0658 | 0.935 (0.503, 1.737) | 0.8305 | 1.144 (0.549, 2.383) | 0.7188 |
| Spring 2021 | 1.109 (0.626, 1.963) | 0.7234 | 1.675 (0.782, 3.59) | 0.1846 | 1.003 (0.335, 3.001) | 0.9954 | 0.684 (0.147, 3.186) | 0.6282 |

[1]eGFR was log transformed to reduce data skewness

[2]Winter 2020–21 served as the reference

Abbreviations: ASCVD, atherosclerotic cardiovascular disease; BMI, body mass index; CCI, Charlson comorbidity index; DBP, diastolic blood pressure; eGFR, estimated glomerular filtration rate; HbA1c, hemoglobin A1c; LDL, low density lipoprotein cholesterol; RAASi, renin angiotensin aldosterone system inhibitor; SBP, systolic blood pressure

This study did not include other cardiometabolic medications, such as anticoagulants, glucagon-like peptide 1 (GLP1) agonists and sodium-glucose cotransporter 2 (SGLT2) inhibitors, that may impact outcomes of COVID-19 [35, 40]. This was an observational study rather than a randomized controlled trial, and therefore causation cannot be established. Data analyzed in

the study were collected in the course of patient care delivery rather than specifically for the study, and therefore testing procedures may not have been uniform among study patients. Information on the patients' SARS-CoV-2 viral load was not available for analysis. Many patients in the study were taking multiple medications, which could have led to drug-drug interactions affecting patient outcomes. Lastly, we were unable to examine biomarkers that may have highlighted other mechanisms that could explain the associations observed.

## Conclusions

This study did not find a consistent evidence of benefit of cardioprotective medications for patients with COVID-19. However, it is important to note that even with increased availability of COVID-19 vaccines, elderly and immunocompromised patients with breakthrough COVID-19 infections remain at risk for severe adverse outcomes. Effective treatments (e.g. monoclonal antibodies and antiviral drugs that are already used for patient care as well as therapies under development, such as DNA / RNA aptamers) are emerging but are not yet universally available; their availability may remain limited in resource-constrained settings and / or emerging economies. We should therefore continue to rigorously assess whether cheap and universally available medications, like the ones analyzed in this study, could benefit patients with COVID-19.

## Author Contributions

**Conceptualization:** Alexander Turchin.

**Data curation:** Fritha J. Morrison.

**Formal analysis:** Fritha J. Morrison, Maxwell Su.

**Funding acquisition:** Alexander Turchin.

**Project administration:** Alexander Turchin.

**Supervision:** Alexander Turchin.

**Writing – original draft:** Fritha J. Morrison.

**Writing – review & editing:** Maxwell Su, Alexander Turchin.

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
