## [Decision Letter · Decision Letter 0]

26 May 2022

PONE-D-22-07504COVID-19 Outcomes in Patients Taking Cardioprotective MedicationsPLOS ONE

Dear Dr. Turchin,

Thank you for submitting your manuscript to PLOS ONE. After careful consideration, we feel that it has merit but does not fully meet PLOS ONE’s publication criteria as it currently stands. Therefore, we invite you to submit a revised version of the manuscript that addresses the points raised during the review process.

The manuscript by Turchin et al. has been evaluated by two Reviewers and should be revised according to the Reviewers' comments. Carefully read the suggestions and respond them appropriately.

We look forward to receiving your revised manuscript.

Kind regards,

Masaki Mogi

Academic Editor

PLOS ONE

Journal Requirements:

4. Thank you for stating the following in the Competing Interests/Financial Disclosure* (delete as necessary) section:

“AT has received research funding from Astra Zeneca, Edwards, Eli Lilly, Novo Nordisk and Sanofi; has equity in Brio Systems; and has served as a consultant for Covance and Proteomics International. None of the other authors have any competing interests.”

We note that one or more of the authors are employed by a commercial company: name of commercial company.

Reviewers' comments:

Reviewer's Responses to Questions

**Comments to the Author**

1. Is the manuscript technically sound, and do the data support the conclusions?

Reviewer #1: Yes

Reviewer #2: Partly

2. Has the statistical analysis been performed appropriately and rigorously? 

Reviewer #1: Yes

Reviewer #2: N/A

3. Have the authors made all data underlying the findings in their manuscript fully available?

Reviewer #1: Yes

Reviewer #2: Yes

4. Is the manuscript presented in an intelligible fashion and written in standard English?

Reviewer #1: Yes

Reviewer #2: No

5. Review Comments to the Author

Reviewer #1: Dr. Alexander Turchin et al conducted a retrospective observational study with a large number of COVID19 patients. They investigated whether patients taking four classes of cardioprotective medications - aspirin, metformin, renin angiotensin aldosterone system inhibitors (RAASi) and statins – have a lower risk of adverse outcomes of COVID-19, and showed lower mortality in patients taking metformin, RAASi, or statins in comparison with those not taking them. The manuscript is well written, and provides an important contribution. I have only few comments on their manuscript.

As already mentioned by the authors, my main concern is the influence of confounders. The patients who discontinued taking medication have usually poor compliance with medications and healthy lifestyle.

In the Discussion section, the authors mention that the mechanism of benefit from RAASi in COVID19 is based on the reduction in ACE2. With my understanding, RAASi does not decrease the expression of ACE2. Furthermore, in the beginning of COVID19 pandemic, it was suggested that RAASi might increase ACE2 and increased ACE2 expression by preexisting RAASi treatment may affect the virus susceptibility. Later, this hypothesis have been rejected. The mechanism of the benefit of RAASi in COVID19 is thought to be derived from anti-inflammatory effects. COVID-19 could cause the imbalanced RAAS and drugs of ACE inhibitors and ARBs balancing RAAS may have the potential benefit on the lung protection in COVID-19.

Reviewer #2: Morrison et al have conducted a retrospective cohort study analysing primary care patients (n=13,585) at a single healthcare delivery system who had a positive reverse transcription-polymerase chain reaction [RT-PCR] result for SARS-CoV-2 between March 2020 and March 2021. The main purpose of the study was to assess whether the intake of four classes of cardioprotective medications -aspirin, metformin, renin angiotensin aldosterone system inhibitors (RAASi) and statins– have a lower risk of adverse outcomes of COVID-19. The authors conclude that cardioprotective medications were not associated with a consistent benefit in adult COVID-19 patients, and only the regular intake of aspirin aspirin had a significantly higher risk of hospitalization in both bivariate and multivariable analyses.

Major issues,

1) The conclusions in the text are unfocused on the present data and should be rephrased.

2) The authors should clarify whether RT-PCR was repeated in the same patients and swab performance (collection timing, procedure, and method of transport) was the same for all patients.

3) The authors should add information regarding magnitude of viral load, medications and outcome of patients.

4) Previous studies have described a relationship between comedications (instead of single medication) and outcome of frail patients (please see Heart Fail Rev. 2021; 26(2): 371–380, GeroScience. 2020 Aug; 42(4): 1021–1049). The authors should mention the above studies and discuss their results in the light of them.

5) The authors should clarify the relationship between nasopharyngeal SARS-CoV-2 viral load at first patient's hospital evaluation and outcome of COVID19 patients. Evidences on this issue are controversial. Previous study has demonstrated that nasopharyngeal SARS-CoV-2 viral load on admission is generally high in patients with COVID-19, regardless of illness severity, but it cannot be used as an independent predictor of unfavorable clinical outcome (please see Sci Rep. 2021 Jun 21;11(1):12931), but other study showed that initial viral load is an incremental predictor of mortality (Mayo Clin Proc Innov Qual Outcomes. 2021 Oct;5(5):891-897).

6) In the light of recent report, extra caution is a d vis e d when reviewing prescriptions of individuals with significant polypharmacy or with renal/hepatic impairment (Clin Pharmacol Ther. 2022 May 14.doi: 10.1002/cpt.2646.). Therefore, the authors should add a perspective regarding drug-durg interactions in COVID19 patients with significant polypharmacy or with cardiac/renal/hepatic impairment.

7) Background should be improved. Therefore, the authors should discuss their results in the light of the following unmentioned studies (please see Eur J Epidemiol. 2022 Feb;37(2):157-165; Clin Res Cardiol. 2021 Jul;110(7):1041-1050; Am J Hypertens. 2022 May 10;35(5):462-469; Metabolism

. 2022 Jun;131:155196. ). What about direct oral anticoagulants or vitamin-K antagonists or antiplatelet therapy or steroids or angiotensin II receptor blockers or other anti-diabetic drugs? What type of statins? Indeed, recent unmentioned RCT demonstrated that atorvastatin increased hospitalization days and imposed negative effects on symptom improvement in hospitalized patients with COVID-19 (J Med Virol. 2022 Jul;94(7):3160-3168.)

8) The authors mention potential future anti-COVID19 drugs (monoclonal antibody treatments and antiviral drugs). However, emerging evidences on DNA/RNA aptamers anti-ACE2 (Pharmacol Res. 2022 Jan;175:105982.) or anti-receptor binding domain of SARS-CoV-2 spike protein (Proc Natl Acad Sci U S A. 2021 Dec 14;118(50):e2112942118. ) promise new development and should be mentioned by the authors.

6. PLOS authors have the option to publish the peer review history of their article (what does this mean?). If published, this will include your full peer review and any attached files.

Reviewer #1: **Yes: **Yasushi Matsuzawa

Reviewer #2: No

---

## [Author Response · Author response to Decision Letter 0]

8 Sep 2022

EDITOR

We note that you have included the phrase “data not shown” in your manuscript. Unfortunately, this does not meet our data sharing requirements. PLOS does not permit references to inaccessible data. We require that authors provide all relevant data within the paper, Supporting Information files, or in an acceptable, public repository.

RESPONSE

We have added Table 4 that includes the data that was previously not shown as advised by the Editor.

Please include your full ethics statement in the ‘Methods’ section of your manuscript file. In

your statement, please include the full name of the IRB or ethics committee who approved or

waived your study, as well as whether or not you obtained informed written or verbal consent.

If consent was waived for your study, please include this information in your statement as

well.

RESPONSE

We have included a full ethics statement in the Methods section (at the end of the Study Cohort subsection).

Thank you for stating the following in the Competing Interests/Financial Disclosure* (delete as necessary) section: “AT has received research funding from Astra Zeneca, Edwards, Eli Lilly, Novo Nordisk and Sanofi; has equity in Brio Systems; and has served as a consultant for Covance and Proteomics International. None of the other authors have any competing interests.” We note that one or more of the authors are employed by a commercial company: name of commercial company. 

1. Please provide an amended Funding Statement declaring this commercial affiliation, as well as a statement regarding the Role of Funders in your study. If the funding organization did not play a role in the study design, data collection and analysis, decision to publish, or

preparation of the manuscript and only provided financial support in the form of authors'

salaries and/or research materials, please review your statements relating to the author

contributions, and ensure you have specifically and accurately indicated the role(s) that these

authors had in your study.

RESPONSE

We wanted to clarify that Phase V Technologies did not provide any financial support for the study. We would like to amend our Competing Interests statement to read as follows:

AT has received research funding from Astra Zeneca, Edwards, Eli Lilly, Novo Nordisk and Sanofi; has equity in Brio Systems; and has served as a consultant for Covance and Proteomics International. MS is an employee of Phase V Technologies. This does not alter our adherence to PLOS ONE policies on sharing data and materials. None of the other authors have any competing interests.

Please also include the following statement within your amended Funding Statement. “The funder provided support in the form of salaries for authors [insert relevant initials], but did not have any additional role in the study design, data collection and analysis, decision to publish, or preparation of the manuscript.”

RESPONSE

We wanted to clarify that Phase V Technologies did not provide any financial support for the study. We would like to amend our Financial Disclosure Statement to read as follows:

This research was funded in part by contract # ME-2019C1-15328 from Patient-Centered Outcomes Research Institute (http://www.pcori.org). The funder only provided financial support in the form of the authors’ (FJM, MS, AT) salaries and research materials and did not play any role in study design, data collection and analysis, decision to publish or preparation of the manuscript. Phase V Technologies did not provide any financial support for the study and did not play any role in study design, data collection and analysis, decision to publish or preparation of the manuscript. The specific roles of the study authors are articulated in the ‘author contributions’ section.

2. Please also provide an updated Competing Interests Statement declaring this commercial

affiliation along with any other relevant declarations relating to employment, consultancy,

patents, products in development, or marketed products, etc.

Within your Competing Interests Statement, please confirm that this commercial affiliation

does not alter your adherence to all PLOS ONE policies on sharing data and materials by

including the following statement: "This does not alter our adherence to PLOS ONE policies

on sharing data and materials.” (as detailed online in our guide for authors

http://journals.plos.org/plosone/s/competing-interests). If this adherence statement is not

accurate and there are restrictions on sharing of data and/or materials, please state these.

Please note that we cannot proceed with consideration of your article until this information has

been declared.

RESPONSE

We would like to amend the Competing Interests statement as outlined above.

REVIEWER # 1

In the Discussion section, the authors mention that the mechanism of benefit from RAASi in COVID19 is based on the reduction in ACE2. With my understanding, RAASi does not decrease the expression of ACE2. Furthermore, in the beginning of COVID19 pandemic, it was suggested that RAASi might increase ACE2 and increased ACE2 expression by preexisting RAASi treatment may affect the virus susceptibility. Later, this hypothesis has been rejected. The mechanism of the benefit of RAASi in COVID19 is thought to be derived from anti-inflammatory effects. COVID-19 could cause the imbalanced RAAS and drugs of ACE inhibitors and ARBs balancing RAAS may have the potential benefit on the lung protection in COVID-19.

RESPONSE

We appreciate the Reviewer’s suggestion and have made changes in both Introduction and Discussion sections in accordance with the Reviewer’s recommendation.

REVIEWER # 2

The conclusions in the text are unfocused on the present data and should be rephrased.

RESPONSE

We have amended the Conclusions section as recommended by the Reviewer.

The authors should clarify whether RT-PCR was repeated in the same patients and swab

performance (collection timing, procedure, and method of transport) was the same for all

patients.

RESPONSE

We would like to clarify that this a real-world evidence study, and therefore all test results that were analyzed were performed for patient care, and not specifically for this study. Consequently it is unlikely that testing procedures have been exactly the same in all patients. We have included this information in the Limitations section of the paper.

The authors should add information regarding magnitude of viral load, medications and

outcome of patients.

RESPONSE

We regret that the information about the viral load was not available for analysis. We have included this in the Limitations section of the manuscript. Multiple patient outcomes (hospitalization, ICU admission, artificial ventilation and death) are already included in the analysis.

Previous studies have described a relationship between comedications (instead of single medication) and outcome of frail patients (please see Heart Fail Rev. 2021; 26(2): 371–380, GeroScience. 2020 Aug; 42(4): 1021–1049). The authors should mention the above studies and discuss their results in the light of them.

RESPONSE

We have included a discussion of and references to the papers recommended by the Reviewer.

The authors should clarify the relationship between nasopharyngeal SARS-CoV-2 viral load at first patient's hospital evaluation and outcome of COVID19 patients.

RESPONSE

We regret that the information on nasopharyngeal SARS-CoV-2 viral load was not available for analysis. We have included this in the Limitations section of the manuscript.

In the light of recent report, extra caution is advised when reviewing prescriptions of individuals with significant polypharmacy or with renal/hepatic impairment (Clin Pharmacol Ther. 2022 May 14.doi: 10.1002/cpt.2646.). Therefore, the authors should add a perspective regarding drug-drug interactions in COVID19 patients with significant polypharmacy or with cardiac/renal/hepatic impairment.

RESPONSE

We have included this in the Limitations section of the manuscript.

Background should be improved. Therefore, the authors should discuss their results in the light of the following unmentioned studies (please see Eur J Epidemiol. 2022 Feb;37(2):157-165; Clin Res Cardiol. 2021 Jul;110(7):1041-1050; Am J Hypertens. 2022 May 10;35(5):462- 469; Metabolism . 2022 Jun;131:155196. ). What about direct oral anticoagulants or vitamin-K antagonists or antiplatelet therapy or steroids or angiotensin II receptor blockers or other anti-diabetic drugs? What type of statins? Indeed, recent unmentioned RCT demonstrated that atorvastatin increased hospitalization days and imposed negative effects on symptom improvement in hospitalized patients with COVID-19 (J Med Virol. 2022 Jul;94(7):3160-3168.)

RESPONSE

We appreciate the Reviewer’s suggestion and have included discussion of all of the studies referenced above in the Introduction and / or Discussion sections of the manuscript.

The authors mention potential future anti-COVID19 drugs (monoclonal antibody treatments and antiviral drugs). However, emerging evidences on DNA/RNA aptamers anti-ACE2 (Pharmacol Res. 2022 Jan;175:105982.) or anti-receptor binding domain of SARS-CoV-2 spike protein (Proc Natl Acad Sci U S A. 2021 Dec 14;118(50):e2112942118. ) promise new development and should be mentioned by the authors.

RESPONSE

We have included information on DNA / RNA aptamers being developed for treatment of COVID in the manuscript as advised by the Reviewer.

Thank you for your thoughtful feedback, and we appreciate the opportunity to revise our manuscript. Please do not hesitate to contact us should you have any further questions.

Yours sincerely,

Alexander Turchin, MD, MS

---

## [Decision Letter · Decision Letter 1]

13 Sep 2022

PONE-D-22-07504R1COVID-19 Outcomes in Patients Taking Cardioprotective MedicationsPLOS ONE

Dear Dr. Turchin,

Thank you for submitting your manuscript to PLOS ONE. After careful consideration, we feel that it has merit but does not fully meet PLOS ONE’s publication criteria as it currently stands. Therefore, we invite you to submit a revised version of the manuscript that addresses the points raised during the review process.

Minor revisions are necessary for the present form. See the comments.

We look forward to receiving your revised manuscript.

Kind regards,

Masaki Mogi

Academic Editor

PLOS ONE

Journal Requirements:

Reviewers' comments:

Reviewer's Responses to Questions

**Comments to the Author**

1. If the authors have adequately addressed your comments raised in a previous round of review and you feel that this manuscript is now acceptable for publication, you may indicate that here to bypass the “Comments to the Author” section, enter your conflict of interest statement in the “Confidential to Editor” section, and submit your "Accept" recommendation.

Reviewer #1: All comments have been addressed

Reviewer #2: (No Response)

2. Is the manuscript technically sound, and do the data support the conclusions?

Reviewer #1: Yes

Reviewer #2: Partly

3. Has the statistical analysis been performed appropriately and rigorously? 

Reviewer #1: Yes

Reviewer #2: I Don't Know

4. Have the authors made all data underlying the findings in their manuscript fully available?

Reviewer #1: Yes

Reviewer #2: Yes

5. Is the manuscript presented in an intelligible fashion and written in standard English?

Reviewer #1: Yes

Reviewer #2: Yes

6. Review Comments to the Author

Reviewer #1: The authors have well revised their manuscript according to my comments. I do not have any more comments on it.

Reviewer #2: The authors partially answered the questions put forward by this reviewer. However, the suggestion on the use of DNA/RNA aptamers remains. The authors should better mention and discuss perspective of aptamers directed toward ACE2 (Pharmacol Res. 2022 Jan;175:105982.) and those directed toward the coronavirus spike protein (Proc Natl Acad Sci U S A. 2021 Dec 14;118(50):e2112942118.).

7. PLOS authors have the option to publish the peer review history of their article (what does this mean?). If published, this will include your full peer review and any attached files.

Reviewer #1: **Yes: **Yasushi Matsuzawa

Reviewer #2: No

---

## [Author Response · Author response to Decision Letter 1]

16 Sep 2022

The authors partially answered the questions put forward by this reviewer. However, the suggestion on the use of DNA/RNA aptamers remains. The authors should better mention and discuss perspective of aptamers directed toward ACE2 (Pharmacol Res. 2022 Jan;175:105982.) and those directed toward the coronavirus spike protein (Proc Natl Acad Sci U S A. 2021 Dec 14;118(50):e2112942118.).

RESPONSE

We have added a discussion of DNA / RNA aptamers being developed for treatment of COVID in the manuscript (in the Discussion section) as advised by the Reviewer (including the references recommended by the Reviewer).

---

## [Decision Letter · Decision Letter 2]

26 Sep 2022

COVID-19 Outcomes in Patients Taking Cardioprotective Medications

PONE-D-22-07504R2

Dear Dr. Turchin,

We’re pleased to inform you that your manuscript has been judged scientifically suitable for publication and will be formally accepted for publication once it meets all outstanding technical requirements.

Kind regards,

Masaki Mogi

Academic Editor

PLOS ONE

Additional Editor Comments (optional):

Reviewers' comments:

Reviewer's Responses to Questions

**Comments to the Author**

1. If the authors have adequately addressed your comments raised in a previous round of review and you feel that this manuscript is now acceptable for publication, you may indicate that here to bypass the “Comments to the Author” section, enter your conflict of interest statement in the “Confidential to Editor” section, and submit your "Accept" recommendation.

Reviewer #2: All comments have been addressed

2. Is the manuscript technically sound, and do the data support the conclusions?

Reviewer #2: Yes

3. Has the statistical analysis been performed appropriately and rigorously? 

Reviewer #2: Yes

4. Have the authors made all data underlying the findings in their manuscript fully available?

Reviewer #2: Yes

5. Is the manuscript presented in an intelligible fashion and written in standard English?

Reviewer #2: Yes

6. Review Comments to the Author

Reviewer #2: (No Response)

7. PLOS authors have the option to publish the peer review history of their article (what does this mean?). If published, this will include your full peer review and any attached files.

Reviewer #2: No

---

## [Editor Report · Acceptance letter]

29 Sep 2022

PONE-D-22-07504R2 

COVID-19 Outcomes in Patients Taking Cardioprotective Medications 

Dear Dr. Turchin:

I'm pleased to inform you that your manuscript has been deemed suitable for publication in PLOS ONE. Congratulations! Your manuscript is now with our production department. 

Kind regards, 

on behalf of

Dr. Masaki Mogi 

Academic Editor

PLOS ONE